# Sampling from Binary Quadratic Distributions via Stochastic Localization

**Chenguang Wang** [* 1 2]  **Kaiyuan Cui** [* 3]  **Weichen Zhao** [4]  **Tianshu Yu** [1]

## Abstract

Sampling from binary quadratic distributions (BQDs) is a fundamental but challenging problem in discrete optimization and probabilistic inference. Previous work established theoretical guarantees for stochastic localization (SL) in continuous domains, where MCMC methods efficiently estimate the required posterior expectations during SL iterations. However, achieving similar convergence guarantees for discrete MCMC samplers in posterior estimation presents unique theoretical challenges. In this work, we present the first application of SL to general BQDs, proving that after a certain number of iterations, the external field of posterior distributions constructed by SL tends to infinity almost everywhere, hence satisfy Poincaré inequalities with probability near to 1, leading to polynomial-time mixing. This theoretical breakthrough enables efficient sampling from general BQDs, even those that may not originally possess fast mixing properties. Furthermore, our analysis, covering enormous discrete MCMC samplers based on Glauber dynamics and Metropolis-Hastings algorithms, demonstrates the broad applicability of our theoretical framework. Experiments on instances with quadratic unconstrained binary objectives, including maximum independent set, maximum cut, and maximum clique problems, demonstrate consistent improvements in sampling efficiency across different discrete MCMC samplers.

---
*Equal contribution [1]School of Data Science, The Chinese University of Hong Kong, Shenzhen [2]Shenzhen Research Institute of Big Data [3]The Academy of Mathematics and Systems Science, Chinese Academy of Sciences [4]The School of Statistics and Data Science, LPMC & KLMDASR, Nankai University. Correspondence to: Weichen Zhao <zhaoweichen@nankai.edu.cn>, Tianshu Yu <yutianshu@cuhk.edu.cn>.

*Proceedings of the 42nd International Conference on Machine Learning*, Vancouver, Canada. PMLR 267, 2025. Copyright 2025 by the author(s).

## 1. Introduction

In this work, we consider sampling from the Gibbs measure with the form of binary quadratic distributions (BQDs):

$$\nu(x) \propto e^{-\frac{\beta}{2}\langle x, Wx\rangle + \langle x, b\rangle}, \ x \in \{-1, 1\}^N. \tag{1}$$

where $\beta$ is the inverse temperature. This distribution class naturally arises in statistical physics such as Ising models (Nishimori, 2001; Bauerschmidt & Dagallier, 2024), spin systems (Bauerschmidt & Bodineau, 2019) and in combinatorial optimization as quadratic unconstrained binary optimization (QUBO) problems (Lucas, 2014; Glover et al., 2022a;b), making it a fundamental object of study in both theoretical and applied research. Sampling from such distributions poses significant challenges due to their discrete nature and complex dependencies (Sly & Sun, 2012). To sample from such distributions, various sampling approaches have been proposed, including Markov Chain Monte Carlo (MCMC) methods (Metropolis et al., 1953; Titsias & Yau, 2017; Grathwohl et al., 2021; Sun et al., 2021; 2023b), simulated annealing (Kirkpatrick et al., 1983; Sun et al., 2022; Sanokowski et al., 2023), diffusion models (Sanokowski et al., 2024) and variational inference (Koehler et al., 2022).

Recently, stochastic localization (SL) has emerged as a promising sampling framework. In the continuous domain, SLIPS (Grenioux et al., 2024) established theoretical guarantees by proving the "duality of log-concavity" on sampling from the posterior and marginal distributions during SL iterations under certain assumptions, leading to significant improvements in sampling efficiency using Metropolis-Adjusted Langevin Algorithm (MALA) (Roberts & Tweedie, 1996). Intuitively, rather than directly sampling from the continuous target distribution, SL achieves sampling by iteratively adding Gaussian noise to posterior estimates of the target distribution. This results in posterior distributions that are Gaussian convolutions of the target distribution which effectively reduces sampling complexity by smoothing out the target distribution's irregular features (Huang et al., 2023; Chen et al., 2024). This idea of decomposing a difficult direct sampling task into a sequence of sampling from simpler distributions is also prevalent in diffusion models for generative tasks and is considered one of the key factors behind their remarkable performance (Ho et al., 2020; Song et al., 2020a;b; Rombach et al., 2022).

However, this elegant theoretical machinery relies heavily on properties specific to continuous spaces and cannot be directly extended to discrete settings, where the fundamental nature of the state space requires different analytical approaches. While several pioneering works have explored SL-based sampling for specific discrete statistical physics models, including Sherrington-Kirkpatrick (SK) models (El Alaoui et al., 2022) and random field Ising models (Alaoui et al., 2023b), these efforts have often employed model-specific techniques for posterior estimation or focused on distributions with particular structural properties. For instance, El Alaoui et al. (2022) addresses SK models at specific temperature regimes using TAP free energy (Nishimori, 2001) optimization to approximate posterior expectations in SL iterations. Consequently, a general framework for leveraging SL with standard, off-the-shelf discrete MCMC samplers and providing theoretical guarantees broadly applicable to the rich class of BQDs has been less explored. A natural question thus arises:

*Can SL reduce sampling difficulty for BQDs, as it does in continuous settings, by constructing easily-samplable posterior distributions using general-purpose discrete MCMC?*

In this work, we affirmatively answer the question for general BQDs by establishing strong theoretical guarantees. Our core theoretical contribution is proving that after a certain number of SL iterations, the posterior distributions constructed by SL satisfy Poincaré inequalities with probability close to 1. This result is crucial as it implies polynomial-time mixing for any underlying discrete MCMC sampler applied to these posteriors. We explicitly derive a bound on the required number of SL iterations, providing concrete guidance for practical implementation. Unlike prior results tied to specific cases, our framework applies to general BQDs (for arbitrary $W$ and $b$), demonstrating the broad applicability of SL in this discrete setting. Importantly, our guarantees hold for discrete MCMC samplers based on both Glauber dynamics and Metropolis-Hastings algorithms, encompassing most commonly used discrete sampling methods.

To demonstrate the practical value of our theoretical insights, we conduct extensive experiments on three types of QUBO problems: maximum independent set, maximum cut, and maximum clique problems, each evaluated across multiple datasets. We assess three representative discrete MCMC samplers based on Glauber dynamics and Metropolis-Hastings algorithms: Gibbs with Gradients (GWG) (Grathwohl et al., 2021), Path Auxiliary Sampler (PAS) (Sun et al., 2021), and Discrete Metropolis-Adjusted Langevin Algorithm (DMALA) (Zhang et al., 2022). Our experimental results demonstrate that SL consistently improves the sampling performance of all three samplers across every dataset, providing strong empirical support for our theoretical guarantees.

## 2. Preliminaries

In this section, we introduce key concepts and notations that will be used throughout this paper.

### 2.1. Notations

Throughout this paper, we consider a system of size $N$ where $[N] := 1, 2, \ldots, N$ denotes the index set and $\Omega := \{-1, 1\}^N$ represents the hypercube state space. For any configuration $x$, we use $x_{\sim i}$ to denote $x$ with site $i$ excluded, $x^i$ to denote the configuration after transition at site $i$ and $x = y$ off j to denote $x_i = y_i, i \neq j$. We define the trivial metric as $d(x, y) := \mathbf{1}_{x \neq y}$, where $\mathbf{1}_{x \neq y} = 1$ if $x \neq y$, and $\mathbf{1}_{x \neq y} = 0$ if $x = y$. We use $|| \cdot ||_{TV}$ to denote the total variation distance. The conditional distribution of $x_i$ is denoted by $\nu_i(\cdot | x)$, while $\mathbb{E}_\nu$ represents expectation under measure $\nu$. Finally, $\langle \cdot, \cdot \rangle$ denotes the standard inner product.

### 2.2. Discrete MCMC Samplers

In this part, we introduce three recent and effective discrete MCMC samplers, which will facilitate our theoretical analysis in Section 4: Gibbs With Gradients (GWG) (Grathwohl et al., 2021), Path Auxiliary Sampler (PAS) (Sun et al., 2021), and Discrete Metropolis-adjusted Langevin Algorithm (DMALA) (Zhang et al., 2022). These samplers share a common foundation in locally balanced proposals, where they incorporate local information about the target distribution to construct more efficient transitions. Essentially, they can all be viewed as generalized Glauber dynamics that incorporate gradient-based information from target distribution and different dimension update schemes, followed by an optional Metropolis-Hastings (MH) adjustment.

For GWG, the locally informed proposal is defined as:

$$p(x'|x) \propto e^{\frac{d(x)}{2}} \mathbf{1}_{x' \in H(x)}, \qquad (2)$$

where $d(x) = -(2x - 1)\nabla \log \nu(x)$, $H(x)$ denotes the Hamming ball of size 1 around $x$, and the site to flip is determined by sampling $i \sim \text{Categorical}(\text{Softmax}(\frac{d(x)}{2}))$.

For PAS, the proposal is formulated as:

$$p_y(x'|x) \propto g(e^{\langle \nabla \log \nu(y), x'-x \rangle}), \qquad (3)$$

where $g(\cdot)$ is the local balancing function (specifically, $g(t) = \sqrt{t}$ in PAS) and $y$ represents the current state as the starting point of the auxiliary path.

For DMALA, the proposal admits the conditional independence relationship $p(x'|x) = \prod_i^N p(x_i'|x)$ and takes the form:

$$p(x'|x) = \prod_{i=1}^N \left( \text{Softmax} \left( \frac{1}{2} (\nabla \log \nu(x))_i (x_i' - x_i) \right) \right), \qquad (4)$$

where the quadratic term present in the original form becomes negligible in the binary case.

Each of these samplers may include an optional MH adjustment step with the acceptance ratio:

$$\min\left(1, \frac{\nu(x')p(x|x')}{\nu(x)p(x'|x)}\right). \tag{5}$$

## 2.3. SL for Sampling

El Alaoui & Montanari (2022) established a connection between stochastic localization and information theory through an *observation process* $(Y_t)_{t\geq 0}$:

$$Y_t = tX + B_t, \tag{6}$$

where $X \sim \nu$, $(B_t)_{t\geq 0}$ is a standard Brownian motion independent of $X$. The conditional distribution $q_t$ of $X$ given $Y_t$ follows a stochastic localization process, which by Theorem 7.12 of (Liptser & Shiryaev, 2013), satisfies:

$$dY_t = u_t(Y_t)dt + dB_t, \quad Y_0 = 0, \tag{7}$$

where $u_t(y) = \int x q_t(x|y)dx$. Grenioux et al. (2024) later generalized this to:

$$Y_t^\alpha = \alpha(t)X + \sigma B_t, \quad t \in [0, T_{\text{gen}}), \tag{8}$$

with $\alpha(t) = t^{1/2}g(t)$ satisfying certain regularity conditions[1]. The corresponding SDE becomes:

$$dY_t^\alpha = \dot{\alpha}(t)u_t^\alpha(Y_t^\alpha)dt + \sigma dB_t, \quad Y_0^\alpha = 0, \tag{9}$$

where $u_t^\alpha(y) = \int x q_t^\alpha(x|y)dx$. We refer more fundamental definitions of SL to Appendix B.1.

## 3. Binary Quadratic Sampling via Stochastic Localization

In this section, we present the SL-based sampling framework for binary quadratic distributions. Based on SL framework introduced in Section 2.3, estimating the posterior expectation $u_t^\alpha(y) = \int x q_t^\alpha(x|y)dx$ is crucial for sampling. The most straightforward approach is to approximate this expectation through Monte Carlo approximation using discrete MCMC samplers:

$$\tilde{U}_t^\alpha = \frac{1}{n}\sum_{j=1}^{n} x_j^t \approx u_t^\alpha(y)$$

where $\{x_j^t\}_{j=1}^n \sim q_t^\alpha(x|y)$. The algorithm is detailed in Algorithm 1. Understanding the properties of this posterior distribution becomes central to our analysis.

---

[1] Specifically, $g \in C^0([0, T_{\text{gen}}), \mathbb{R}+) \cap C^1((0, T_{\text{gen}}), \mathbb{R}+)$ is strictly increasing with $g(t) \sim Ct^{\beta/2}$ as $t \to 0$ for some $\beta \geq 1, C > 0$, and $\lim_{t \to T_{\text{gen}}} g(t) = \infty$.

---

**Algorithm 1** SL with Discrete MCMC Samplers

**Input:** Discrete MCMC Sampler (DMS), time steps for SDE iterations $\{t_i\}_{i=0}^T$, sample size for posterior estimation $n$, SL parameters $\alpha(\cdot), \sigma$
Initialize $\tilde{Y}_0 \sim \mathcal{N}(\mathbf{0}, t_0\sigma^2 I)$
**for** $i = 0$ **to** $T - 1$ **do**
    Set $\delta_i = t_{i+1} - t_i, w_i = \alpha(t_{i+1}) - \alpha(t_i)$
    Simulate $\{x_j^i\}_{j=1}^n \sim \mathbb{P}(X|\tilde{Y}_i)$ with DMS
    Estimate the posterior expectation $\tilde{U}_i^\alpha = \frac{1}{n}\sum_{j=1}^n x_j^i$
    Simulate $\tilde{Y}_{i+1} \sim \mathcal{N}(\tilde{Y}_i + w_i\tilde{U}_i^\alpha, \sigma^2\delta_i \mathbf{I})$
**end for**
**Output:** $\frac{\tilde{Y}_T}{\alpha(t_T)}$

---

Given the observation process in Equation 8, Bayes' theorem yields the posterior distribution of $X$ given $Y_t$:

$$q_t^\alpha(x|y) \propto p_t^\alpha(y|x)\nu(x) \tag{10}$$

where $p_t^\alpha(y|x)$ is the Gaussian likelihood induced by the observation process. For the binary quadratic distribution $\nu(x)$ defined in Equation (1), the posterior distribution takes the explicit form:

$$q_t^\alpha(x|y) \propto e^{-\frac{\beta}{2}\langle x, Wx\rangle + \langle x, b + \frac{\alpha(t)Y_t}{\sigma^2 t}\rangle}. \tag{11}$$

Note that the quadratic term associated with $x$ from the Gaussian likelihood $p_t^\alpha(y|x)$ vanishes since $\langle x, x\rangle = \mathbf{1}$ for $x \in \{-1, 1\}^N$. In contrast to continuous settings, where $x^2$ terms persist and log-concavity can be analyzed through the Hessian of the log-likelihood, the discrete case presents unique challenges. Here, the posterior distribution differs from $\nu(x)$ only in its linear term, and we cannot assess sampling difficulty through derivatives. This fundamental difference necessitates theoretical guarantees to understand how SL affects the mixing properties of discrete MCMC samplers.

**Physical Intuition** Before delving into rigorous theoretical analysis, we provide an intuitive understanding from a physical perspective. Our starting point is the following convergence result:

**Theorem 3.1.** *Consider the observation process*

$$Y_t = \alpha(t)X + \sigma B_t,$$

*in the case* $\frac{\alpha(t)}{\sigma\sqrt{t}} \to +\infty$ *as* $t \to +\infty$, *for arbitrary* $\zeta > 0$ *and* $\varepsilon > 0$, *there is a* $T(\zeta, \varepsilon)$ *large enough such that for* $t \geq T$, *the observation process satisfies*

$$\mathbb{P}(|Y_t| \geq \zeta) \geq 1 - \varepsilon.$$

Examining the linear term coefficient in the posterior distribution (11), which serves as an effective external field

$h_t$:

$$h_t = b + \frac{\alpha(t)Y_t}{\sigma^2 t}. \qquad (12)$$

The behavior of this term is intrinsically linked to the stochastic localization process. Specifically, the dynamics of the localization variable $Y_t$ are governed by the SL formulation arising from an observation process, as detailed in Section 2.3 (cf. Equation (8)). Theorem 3.1 provides a theoretical guarantee on the asymptotic behavior of this process under our chosen $\alpha(t)$ schedule. It proves that as $t \to T_{\text{gen}}$, the term $\frac{Y_t}{\alpha(t)} \to X \in \{-1, 1\}^N$ and the scaling factor $\frac{\alpha(t)^2}{\sigma^2 t} \to +\infty$. As a result, the magnitude of the effective external field $|h_t|$ grows infinitely. In physical terms, this corresponds to applying an increasingly strong external field to an interacting system. For some statistical physics models, the presence of a strong external field generally enhances mixing properties compared to the zero-field case (Martinelli et al., 1994; Martinelli & Olivieri, 1994; Kerimov, 2012). Intuitively, when this field becomes sufficiently large, the quadratic interaction term becomes negligible in comparison, and the posterior distribution approximates to:

$$q_t^\alpha(x|y) \stackrel{\approx}{\propto} e^{\langle x, b + \frac{\alpha(t)Y_t}{\sigma^2 t} \rangle}.$$

This simplifies to a product of independent Bernoulli distributions, which is significantly easier to sample. This property of a strong, growing external field is crucial for establishing favorable mixing properties of the posterior distribution at late times. A formal condition on the external field strength required for our subsequent theoretical guarantees is introduced later in Condition 4.1.

## 4. Theoretical Analysis

This section establishes theoretical guarantees for sampling from posterior distributions in the SL framework. The spectral gap $\gamma_{\text{gap}}$ determines the convergence rate of algorithm and takes different forms for each DMCMC sampler:

- $1 - \frac{|h|}{e^{\frac{3|h|}{4}} + e^{-\frac{3|h|}{4}}}$ for Glauber dynamics;

- $1 - 2|h|e^{-|h|}$ for classical Metropolis chains;

- $1 - \frac{|h|}{(e^{\frac{|h|}{4}} + e^{-\frac{|h|}{4}})^2}$ for single-site gradient-informed MH algorithms;

- $1 - \frac{4|h|N}{(e^{\frac{|h|}{4}} + e^{-\frac{|h|}{4}})^2}$ for DULA (Zhang et al., 2022).

Notably, for all cases, $\gamma_{\text{gap}}$ increases with $|h|$ and approaches 1 as $|h| \to +\infty$.

We organize this section as follows. Section 4.1 introduces the essential definitions and assumptions that underpin our analysis. Section 4.2 examines two warm-up cases, establishing Poincaré inequalities for classical Glauber dynamics and Metropolis-Hastings algorithms. Building on these foundations, Section 4.3 extends our theoretical guarantees to more sophisticated sampling algorithms. All proofs are deferred to Appendix C.

### 4.1. Setup and Key Assumptions

For simplicity, we represent the posterior distribution in equation (11) using the following Gibbs measure:

$$\nu_{\beta, h} = \frac{1}{\tilde{Z}_{\beta, h}} e^{-\frac{\beta}{2}\langle x, Wx \rangle + \langle x, h \rangle}, x \in \{-1, 1\}^N, \qquad (13)$$

where $\beta$ denotes the inverse temperature and $\tilde{Z}_{\beta, h}$ is the partition function. Based on the physical intuition provided in Section 3, the external field $h$ in equation (13) grows to infinity, which motivates our fundamental assumption for all subsequent theoretical analysis:

**Condition 4.1.** The strong external field $h$ satisfies

$$|h| \geq 2\beta \sup_{i \in [N]} \sum_{k \neq i} |W_{ik}|, \qquad (14)$$

*Remark* 4.2. Theorem 3.1 guarantees that this condition holds with high probability after sufficient SL iterations. Moreover, for low-temperature models (large inverse temperature $\beta$), this condition is naturally satisfied due to the dominance of the external field term.

Following Condition 4.1 and Theorem 3.1, setting $\zeta = 2\beta \sup_{i \in [N]} \sum_{k \neq i} |W_{ik}|$ and by choosing a sufficiently small $\varepsilon > 0$, we can ensure that for $t \geq T(\alpha, \varepsilon)$, the external field $|Y_t|$ in equation (11) is larger than $\zeta$, and therefore exceeds $2\beta \sup_{i \in [N]} \sum_{k \neq i} |W_{ik}|$ with probability at least $1 - \varepsilon$. In subsequent sections, we will demonstrate that when the external field is sufficiently large, certain Discrete MCMC samplers for Gibbs distribution (13) satisfy Poincaré inequalities. This theoretical guarantee, combined with the high-probability bound on the external field, establishes that the Poincaré inequalities in Theorem 4.3 and Theorem 4.5 hold with probability close to 1, thereby ensuring polynomial-time mixing[2] for sampling from the posterior distribution (11).

### 4.2. Warm-up Cases: Poincaré Inequality for Glauber Dynamics and Metropolis Chains

In this subsection, we begin with presenting our theoretical results through two fundamental examples, Glauber dynamics and Metropolis chains. Here, we derive explicit coefficients for the Poincaré inequality that depend on the external field $h$.

---

[2]As shown in Theorem 20.6 of (Levin & Peres, 2017), stochastic dynamics satisfying Poincaré inequality exhibit polynomial mixing time.

**Glauber Dynamics** Glauber dynamics, or Gibbs sampling, provides a fundamental approach for sampling from BQDs. The algorithm iteratively updates each spin $i$ by sampling from its conditional distribution:

$$\nu_{\beta,h}(x_i \mid x_{\sim i}) := \frac{e^{-\frac{\beta}{2}\langle x, Wx \rangle + \langle x, h \rangle}}{\sum_{x_i = \pm 1} e^{-\frac{\beta}{2}\langle x, Wx \rangle + \langle x, h \rangle}}.$$

For Glauber dynamics, we establish the following theorem with spectral gap $\gamma_{\text{gap}} = 1 - \frac{|h|}{e^{\frac{3|h|}{4}} + e^{-\frac{3|h|}{4}}}$:

**Theorem 4.3.** *For the Gibbs measure* (13) *satisfying the large field Condition 4.1, the following Poincaré inequality holds:*

$$Var_{\nu_{\beta,h}}(f) \leq \frac{1}{1 - \frac{|h|}{e^{\frac{3|h|}{4}} + e^{-\frac{3|h|}{4}}}} \mathcal{E}_{GD}(f, f), \quad (15)$$

*where $\mathcal{E}_{GD}(f, f)$ is the Dirichlet form* (23) *associated with Glauber dynamics on $L^2(\nu_{\beta,h})$.*

Building upon the established Poincaré inequality in Theorem 4.3, which quantifies the mixing rate of Glauber dynamics, we can further analyze the convergence properties of Monte Carlo estimators. Specifically, the spectral gap $\gamma_{\text{gap}}$ plays a crucial role in deriving concentration inequalities for averages of samples generated by the dynamics. This leads us to the following corollary concerning the Chernoff-type error bound for our Monte Carlo estimator of the posterior mean.

**Corollary 4.4.** *Under condition of theorem 4.3, for a sequence $\{X_1, X_2, \ldots, X_n\}$ sampled from Glauber dynamics, for all $\varepsilon > 0$, we have:*

$$P_q \left[ \left| \frac{1}{n} \sum_{i=1}^{n} X_i - \mathbb{E}_{\nu_{\beta,h}}[X] \right| \geq \varepsilon \right] \leq C_{\gamma_{gap}, n, q} e^{-\frac{n\varepsilon^2 \gamma_{gap}}{c}},$$

*where $X_1 \sim q$ is the initial distribution of samples, the spectral gap is $\gamma_{gap} = 1 - \frac{|h|}{e^{\frac{3|h|}{4}} + e^{-\frac{3|h|}{4}}}$, $c$ is an absolute constant, and $C_{\gamma_{gap}, n, q}$ is a rational function.*

This established error bound demonstrates that the Monte Carlo estimator achieves an exponential error reduction with respect to the sample size $n$.

**Metropolis Chains.** The Metropolis-Hastings (MH) algorithm is a cornerstone of MCMC sampling methods. As a warm-up case, we analyze the simplified Metropolis chains (Martinelli, 1999) where, for each site $i \in [N]$, the transition $x \to x^i$ occurs with probability:

$$P(x^i \mid x) = \min\{1, \frac{\nu_{\beta,h}(x^i)}{\nu_{\beta,h}(x)}\}$$
$$= \min\{1, e^{2\beta x_i \sum_{j \neq i} W_{ij} x_j - 2h_i x_i}\},$$

and remains unchanged with probability: $1 - P(x^i \mid x) = 1 - \min\{1, e^{2\beta x_i \sum_{j \neq i} W_{ij} x_j - 2h_i x_i}\}$. For these Metropolis chains, we establish the following Poincaré inequality:

**Theorem 4.5.** *For the Gibbs measure* (13) *satisfying the large field Condition* (4.1)*, the following Poincaré inequality holds:*

$$Var_{\nu_{\beta,h}}(f) \leq \frac{1}{1 - 2|h|e^{-|h|}} \mathcal{E}_{MH}(f, f), \quad (16)$$

*where $\mathcal{E}_{MH}(f, f)$ is the Dirichlet form* (24) *associated with MH algorithm on $L^2(\nu_{\beta,h})$.*

### 4.3. Poincaré Inequality for Advanced DMCMC Algorithms

While Section 4.2 provides warm-up cases for classical algorithms, here we analyze more sophisticated DMCMC samplers. We propose a general condition that not only encompasses these warm-up cases but also extends to advanced gradient-informed proposals introduced in Section 2.2. Under this natural condition, we establish similar Poincaré inequalities for these sampling methods.

**Single-Site Metropolis-Hastings Algorithms** Consider a MH algorithm that updates one site at a time with transition kernel:

$$P(x^i|x) = \Psi(x^i|x) \min\left\{1, \frac{\nu_{\beta,h}(x^i)\Psi(x|x^i)}{\nu_{\beta,h}(x)\Psi(x^i|x)}\right\}, \quad (17)$$

and remains at the current state with probability: $P(x^i|x) = 1 - \Psi(x^i|x) \min\left\{1, \nu_{\beta,h}(x^i)\Psi(x|x^i)/\nu_{\beta,h}(x)\Psi(x^i|x)\right\}$, where $\Psi$ denotes the proposal distribution. This formulation generalizes both the GWG transition kernel (2) and the PAS kernel (3) in their single-site update variants.

**Theorem 4.6.** *For the MH kernel* (17)*, assume the proposal $\Psi(x^i \mid x)$ is chosen such that $P(x^i \mid x)$ is Lipschitz continuous for all $x = \hat{x}$ off j:*

$$\left| P(x^i \mid x) - P(\hat{x}^i \mid \hat{x}) \right| \leq C_{Lip}(\beta, h)|\beta W_{ij} x_i x_j|,$$

*where $C_{Lip}(\beta, h)$ decreases exponentially to 0 as the external field $|h|$ tends to infinity. Then for the Gibbs measure* (13) *satisfying the large field Condition 4.1, the following Poincaré inequality holds:*

$$Var_{\nu_{\beta,h}}(f) \leq \frac{1}{1 - C_{Lip}(\beta, h)|h|} \mathcal{E}_{MH}(f, f), \quad (18)$$

*where $\mathcal{E}_{MH}(f, f)$ is the Dirichlet form* (24) *associated with MH dynamics on $L^2(\nu_{\beta,h})$.*

*Remark* 4.7. Judging from the above warm-up cases and the calculation examples below, for the measure $\nu_{\beta,h}$ we are considering, the Lipschitz properties of general cases can be expected.

Particularly, for gradient-informed proposals

$$\Psi(x^i \mid x) := \frac{1}{Z_{\beta,h}(x)} e^{\frac{1}{2}\nabla U(x)_i(x^i-x)},$$

where $U(x) = -\frac{\beta}{2}\langle x, Wx \rangle + \langle x, h \rangle$, we establish the following result:

**Theorem 4.8.** *For the Gibbs measure* (13) *satisfying the large field Condition 4.1, the following Poincaré inequality holds for the gradient-informed Metropolis-Hastings algorithm:*

$$Var_{\nu_{\beta,h}}(f) \le \frac{1}{1 - \frac{|h|}{(e^{\frac{|h|}{4}}+e^{-\frac{|h|}{4}})^2}} \mathcal{E}_{MH}(f,f), \qquad (19)$$

*where $\mathcal{E}_{MH}(f,f)$ denotes the Dirichlet form associated with the MH algorithm on $L^2(\nu_{\beta,h})$.*

**DULA**   The Discrete Unadjusted Langevin Algorithm (DULA) performs simultaneous updates across all sites. Its transition kernel takes the form:

$$P(x^{[N]}|x) = \Psi(x^{[N]}|x) = \prod_{i\in[N]}\Psi(x^i|x), \qquad (20)$$

with probability $1 - \Psi(x^{[N]}|x)$ of remaining at the current state. The gradient-informed proposal for each site is given by:

$$\begin{aligned}
\Psi(x^i \mid x) &:= \frac{1}{Z_{\beta,h}(x)} e^{\frac{1}{2}\nabla U(x)_i(x^i-x)} \\
&= \frac{1}{Z_{\beta,h}(x)} e^{-\frac{\beta}{2}\langle x, W(x^i-x)\rangle + \frac{1}{2}\langle x^i-x, h\rangle} \\
&= \frac{1}{Z_{\beta,h}(x)} e^{\beta x_i \sum_{j\neq i} W_{ij}x_j - hx_i}
\end{aligned}$$

which aligns with the proposal (4). For this algorithm, we establish:

**Theorem 4.9.** *For the Gibbs measure* (13) *satisfying the large field Condition 4.1, the following Poincaré inequality holds:*

$$Var_{\nu_{\beta,h}}(f) \le \frac{1}{1 - \frac{4|h|N}{(e^{\frac{|h|}{4}}+e^{-\frac{|h|}{4}})^2}} \mathcal{E}(f,f), \qquad (21)$$

*where*

$$\mathcal{E}(f,f) := \frac{1}{2}\int \Big(f(x) - f(x^{[N]})\Big)^2 \nu_{\beta,h}(\mathrm{d}x)P(\mathrm{d}x^{[N]}|x)$$

*is the Dirichlet form associated with DULA on $L^2(\nu_{\beta,h})$.*

*Remark* 4.10. Similar to Corollary 4.4, for Metropolis chains, single-site gradient-informed MH algorithms, and DULA, the corresponding Chernoff-type error bounds of the Monte Carlo estimator for the posterior mean can also be obtained. These are derived from Theorems 4.5, 4.8, and 4.9, respectively. For a detailed exposition, please refer to Appendix C.8.

## 5. Related Work

Our work connects to two main research areas: discrete MCMC sampling and stochastic localization. Recent advances in discrete MCMC sampling have moved beyond traditional Gibbs sampling (Dai et al., 2020; Wang & Cho, 2019) to gradient-informed methods, including Gibbs With Gradients (Grathwohl et al., 2021), Path Auxiliary Sampler (Sun et al., 2021), and Discrete Langevin Algorithm (Zhang et al., 2022). These methods enhance sampling efficiency by incorporating local structure information into proposal distributions. Stochastic localization (SL) was initially developed for proving measure properties (Eldan, 2013; 2020) and has recently emerged as a powerful sampling framework (El Alaoui et al., 2022; Grenioux et al., 2024). While SL has shown success in continuous domains through SLIPS (Grenioux et al., 2024) and specific discrete models like SK models (El Alaoui et al., 2022), theoretical analysis for general discrete distributions remains challenging due to their inherent structural constraints. For a comprehensive review of related work, please refer to Appendix A.

## 6. Empirical Results

In this section, we empirically validate the sampling performance of the SL framework. Following the DISCS benchmark (Goshvadi et al., 2024) [3], we evaluate on three types of combinatorial optimization problems with the form of binary quadratic distributions: maximum independent set, maximum cut, and maximum clique problems, each containing multiple diverse datasets (14 datasets in total). Detailed information about these problems and datasets is provided in Appendix D.1.

**Experimental Settings**   We employ three advanced discrete MCMC samplers: Gibbs with Gradients (GWG) (Grathwohl et al., 2021), Path Auxiliary Sampler (PAS) (Sun et al., 2021), and Discrete Metropolis-Adjusted Langevin Algorithm (DMALA) (Zhang et al., 2022), all incorporating Glauber dynamics and Metropolis-Hastings algorithms. To align with our theoretical framework, we implement GWG and PAS with single-site updates, while using the Metropolis-Hastings adjusted version of DULA, that is DMALA. For SL implementation, we adopt the GEOM(2,1) $\alpha$ schedule and uniform SDE time discretization from SLIPS (Grenioux et al., 2024). Hyperparameters are tuned through comprehensive search. For fair comparison when estimating posterior expectations, SL uses

---

[3] While MCMC samplers ideally generate i.i.d. samples from the target distribution after mixing, SL converges to a Dirac distribution of a single sample from the target distribution, making effective sample size metrics from DISCS (Goshvadi et al., 2024) unsuitable for performance evaluation. Therefore, we do not compare different sampling algorithms on traditional physics models under this metric.

*Table 1.* Objective values (↑) on MIS benchmarks with different samplers and their SL variants. For each sampler, we compare the original method and its SL counterpart. Bold values indicate better performance between the paired methods.

| | ER-DENSITY | | | | SATLIB |
| | R-0.05 | R-0.10 | R-0.20 | R-0.25 | |
| --- | --- | --- | --- | --- | --- |
| GWG | $104.219 \pm 1.63$ | $61.750 \pm 0.90$ | $34.125 \pm 0.48$ | $27.813 \pm 0.63$ | $419.063 \pm 14.29$ |
| SL-GWG | $\mathbf{104.375 \pm 1.52}$ | $\mathbf{61.938 \pm 0.93}$ | $\mathbf{34.375 \pm 0.74}$ | $\mathbf{28.000 \pm 0.56}$ | $\mathbf{419.165 \pm 14.39}$ |
| PAS | $104.375 \pm 1.64$ | $61.750 \pm 0.94$ | $34.063 \pm 0.61$ | $27.813 \pm 0.46$ | $419.539 \pm 14.39$ |
| SL-PAS | $\mathbf{104.531 \pm 1.48}$ | $\mathbf{61.906 \pm 0.88}$ | $\mathbf{34.375 \pm 0.55}$ | $\mathbf{28.031 \pm 0.59}$ | $\mathbf{419.707 \pm 14.35}$ |
| DMALA | $103.063 \pm 1.62$ | $61.000 \pm 0.83$ | $33.813 \pm 0.68$ | $27.438 \pm 0.56$ | $\mathbf{415.722 \pm 14.51}$ |
| SL-DMALA | $\mathbf{103.594 \pm 1.80}$ | $\mathbf{61.344 \pm 0.96}$ | $\mathbf{34.000 \pm 0.56}$ | $\mathbf{27.750 \pm 0.66}$ | $415.718 \pm 14.38$ |

identical parameters as the corresponding discrete MCMC sampler and maintains the same 10000 MCMC steps in total. Furthermore, motivated by our theoretical results showing that posterior distributions become easier to sample from as SL iterations proceed, the experimental results in Section 6.1 are obtained by exponentially decaying the allocation of 10000 MCMC steps across SL iterations. The impact of this allocation strategy is analyzed in Section 6.2. Detailed experimental settings are provided in the Appendix D.2.

## 6.1. Main Results

In this section, we evaluate the SL framework against three discrete MCMC samplers (GWG, PAS, DMALA) on maximum independent set (MIS), maximum cut (MaxCut), and maximum clique (MaxClique) problems. Following the setting in DISCS (Goshvadi et al., 2024), for MIS, we directly report the objective values, while for MaxCut and MaxClique, we report the optimality gap ($\frac{obj}{baseline} \times 100\%$) where baseline values are obtained from classic solvers collected by DISCS benchmark. The best solution found during the entire sampling process is used for evaluation. To assess sampling performance, we implement SL using each MCMC sampler for posterior estimation, enabling direct comparison:

MCMC sampler *vs.* SL + MCMC sampler

under *identical* MCMC steps.

**Results on MIS**   Table 1 shows the performance comparison on MIS problems across five datasets, including four Erdős-Rényi random graphs with different densities (0.05-0.25) and instances from SATLIB dataset. For each MCMC sampler, our SL variant consistently achieves better or comparable objective values. The improvements are particularly noticeable on sparse graphs (R-0.05 and R-0.10), where SL-PAS outperforms baseline PAS by 0.15% and 0.25% respectively. Among all methods, SL-PAS demonstrates the strongest overall performance, achieving the highest objective values on both random graphs and SATLIB instance.

*Table 2.* The percentage (%) (↑) of the solution provided by DISCS (Goshvadi et al., 2024) on Maxclique benchmarks. Bold values indicate better performance between paired methods.

| | RB | TWITTER |
| --- | --- | --- |
| GWG | $87.509 \pm 6.19$ | $100.000 \pm 0.00$ |
| SL-GWG | $\mathbf{87.598 \pm 6.16}$ | $100.000 \pm 0.00$ |
| PAS | $87.544 \pm 6.15$ | $100.000 \pm 0.00$ |
| SL-PAS | $\mathbf{87.649 \pm 6.14}$ | $100.000 \pm 0.00$ |
| DMALA | $87.231 \pm 6.14$ | $100.000 \pm 0.00$ |
| SL-DMALA | $\mathbf{87.310 \pm 6.15}$ | $100.000 \pm 0.00$ |

**Results on MaxClique**   Table 2 presents results on two MaxClique datasets: RB and TWITTER. On the RB dataset, all SL variants demonstrate consistent improvements over their MCMC counterparts, with SL-PAS achieving the highest optimality gap of 87.649%. For the TWITTER dataset, all methods achieve optimal solutions (100%), indicating its relative simplicity. The improvements on RB, while modest in magnitude, are consistent across all three MCMC samplers, further validating SL's effectiveness in enhancing sampling performance.

**Results on MaxCut**   Table 3 presents results on seven MaxCut datasets, including Erdős-Rényi (ER) graphs, Barabási-Albert (BA) graphs with varying sizes (256-1024 nodes), and OPTSICOM instances. SL variants consistently outperform their MCMC counterparts across most datasets, with particularly notable improvements on larger graphs. SL-PAS achieves the best overall performance, showing improvements of up to 0.045% on BA-512 and 0.038% on BA-1024 compared to baseline PAS. On the OPTSICOM instance, all methods achieve optimal solutions.

The experimental results demonstrate the consistent effectiveness of our SL framework across different combinatorial optimization problems, datasets, and MCMC samplers. It is important to note that the DMCMC baselines employed are themselves highly effective and often perform near optimal-

*Table 3.* The percentage (%) (↑) of the solution provided by DISCS ([Goshvadi et al., 2024](#)) on Maxcut benchmarks. Bold values indicate better performance between paired methods.

| | ER | | | BA | | | OPTSICOM |
| | 256-300 | 512-600 | 1024-1100 | 256-300 | 512-600 | 1024-1100 | |
|---|---|---|---|---|---|---|---|
| GWG | $101.881 \pm 1.76$ | $100.144 \pm 0.12$ | $100.098 \pm 0.14$ | $99.947 \pm 0.06$ | $100.850 \pm 0.48$ | $101.544 \pm 0.38$ | $100.000 \pm 0.00$ |
| SL-GWG | $\mathbf{101.924 \pm 1.76}$ | $\mathbf{100.161 \pm 0.12}$ | $\mathbf{100.101 \pm 0.13}$ | $\mathbf{99.972 \pm 0.06}$ | $\mathbf{100.889 \pm 0.49}$ | $\mathbf{101.562 \pm 0.37}$ | $100.000 \pm 0.00$ |
| PAS | $101.884 \pm 1.76$ | $100.158 \pm 0.12$ | $100.120 \pm 0.13$ | $99.954 \pm 0.05$ | $100.883 \pm 0.48$ | $101.632 \pm 0.37$ | $100.000 \pm 0.00$ |
| SL-PAS | $\mathbf{101.920 \pm 1.76}$ | $\mathbf{100.174 \pm 0.12}$ | $\mathbf{100.123 \pm 0.14}$ | $\mathbf{99.975 \pm 0.05}$ | $\mathbf{100.928 \pm 0.48}$ | $\mathbf{101.670 \pm 0.37}$ | $100.000 \pm 0.00$ |
| DMALA | $101.871 \pm 1.76$ | $100.130 \pm 0.12$ | $100.074 \pm 0.14$ | $99.946 \pm 0.06$ | $100.763 \pm 0.50$ | $\mathbf{101.243 \pm 0.39}$ | $100.000 \pm 0.00$ |
| SL-DMALA | $\mathbf{101.922 \pm 1.76}$ | $\mathbf{100.144 \pm 0.12}$ | $\mathbf{100.078 \pm 0.14}$ | $\mathbf{99.965 \pm 0.06}$ | $\mathbf{100.792 \pm 0.49}$ | $101.228 \pm 0.38$ | $100.000 \pm 0.00$ |

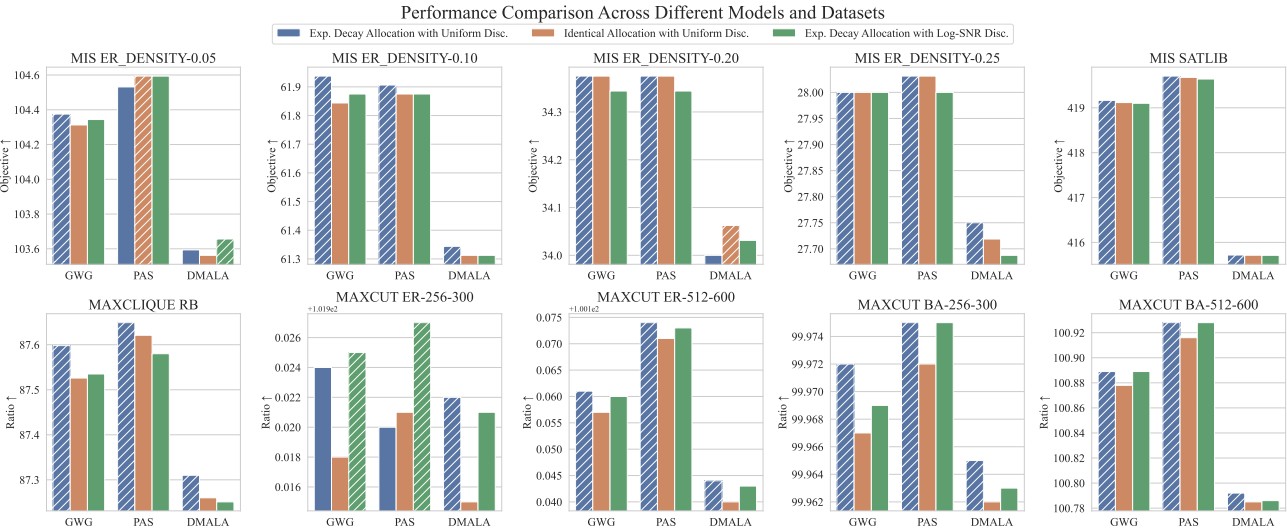

*Figure 1.* Ablation study comparing two design choices: (1) MCMC steps allocation strategies (Exponential Decay vs. Identical) and (2) SDE time discretization methods (Uniform vs. Log-SNR). Hatched bars indicate the best performing configuration for each algorithm-dataset combination.

ity on these challenging tasks, significantly outperforming commercial solvers like Gurobi (as shown in Table 3). The improvements shown by SL are achieved even when starting from these strong established methods, highlighting its ability to further boost performance. For comprehensive evaluation results, including the sampling trajectory, we refer readers to Appendix D.3.

### 6.2. Ablation Study

In this part, we conduct ablation studies on two aspects: (1) MCMC steps allocation strategies (Exponential Decay vs. Identical) and (2) SDE time discretization methods (Uniform vs. Log-SNR), to validate our physical intuition proposed in Section 3 and theoretical insights about sampling difficulty evolution proved in Section 4.

**Time Discretization Strategy**  We compare two strategies for discretizing the SDE integration interval: uniform dis-

cretization (our default choice) and Log-SNR discretization proposed in SLIPS ([Grenioux et al., 2024](#)). Under uniform discretization, time points are evenly spaced across the interval, while Log-SNR discretization allocates more SDE iterations to smaller time points where posterior distributions are theoretically more challenging to sample from.

**Impact of MCMC Step Allocation**  Our theory suggests that sampling becomes easier during SL iterations. To leverage this insight, we compare two strategies for allocating MCMC steps across SL iterations: exponential decay (our default choice) and uniform allocation. In exponential decay, earlier iterations receive more MCMC steps for posterior estimation, while uniform allocation distributes steps equally.

As shown in Figure 1, Exp. Decay Allocation with Uniform Disc. achieves superior performance in most test cases, which aligns well with our physical intuition and theoretical analysis. However, we observe several interesting excep-

tions where alternative strategies perform marginally better, such as in MAXCUT ER-256-300 with PAS algorithm where Exp. Decay Allocation with Log-SNR Disc. yields the best result, and in some MIS SATLIB cases where Identical Allocation shows comparable performance. Despite these minor variations, the performance differences among the three strategies are generally modest (typically less than 1% in objective values or optimality gap), demonstrating the remarkable stability of SL sampling framework across different configurations.

For more comprehensive ablation studies, including the impact of $\alpha$-schedule in SL and the influence of hyperparameters on sampling performance, we refer readers to Appendix D.4, where we provide detailed experimental results and analyses.

## 7. Conclusion and Discussion

In this work, we have established the first theoretical framework for applying stochastic localization (SL) to binary quadratic distributions (BQDs), proving that the posterior distributions constructed by SL satisfy Poincaré inequalities with high probability after a certain number of iterations. Our theoretical guarantees are particularly general, encompassing both Glauber dynamics and Metropolis-Hastings based discrete MCMC samplers without restrictive assumptions on the underlying distributions. Extensive experiments on QUBO problems demonstrate that SL consistently improves the sampling efficiency of various discrete MCMC samplers, providing strong empirical support for our theoretical results.

These results also open several directions for future research. First, extending our framework beyond BQDs to distributions with unknown forms (specifically, deep energy-based models) remains challenging, potentially requiring efficient second-order Taylor approximations. Second, while our current theoretical analysis provides mixing guarantees for the posterior sampling step within SL, establishing rigorous theoretical guarantees for the convergence rate of the overall SL process to the final target distribution is a crucial and challenging problem that warrants future investigation. Third, generalizing our framework to handle discrete variables with more than two states, moving beyond the binary case, is an important direction.

## Acknowledgments

This work was supported by the National Key R&D Program of China under grant 2022YFA1003900, the National Natural Science Foundation of China (No.12401666, 12326611, 12426303), the Guangdong Provincial Key Laboratory of Mathematical Foundations for Artificial Intelligence (2023B1212010001) and the Fundamental Research Funds for the Central Universities, Nankai University (No.054-63241437).

## Impact Statement

This work has significant implications for both theoretical research and practical applications in discrete optimization and probabilistic inference. Our theoretical guarantees for SL in discrete domains open new avenues for developing more efficient sampling algorithms. The demonstrated improvements in sampling efficiency across various discrete MCMC samplers suggest potential applications in diverse fields, including statistical physics, combinatorial optimization, and machine learning. As discrete optimization problems continue to arise in emerging technologies such as quantum computing and molecular design, the practical impact of our theoretical advances is expected to grow significantly.

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

## A. Related Work

**Discrete MCMC Samplers**   The Gibbs sampling algorithm, which is equivalent to Glauber dynamics in statistical physics, has been widely used to study statistical physics models (refs) and train deep energy-based models (Dai et al., 2020; Wang & Cho, 2019). To improve sampling efficiency beyond traditional Gibbs sampling, recent works have focused on locally balanced proposals (Zanella, 2020) that leverage local information such as gradient-like quantities. These methods modify the proposal distribution by incorporating local structure of the target distribution, leading to more informed transitions. Notable examples include Gibbs With Gradients (Grathwohl et al., 2021), which approximates probability ratios using well-defined gradient information in discrete space, Path Auxiliary Sampler (Sun et al., 2021), which constructs auxiliary paths to enable long-range transitions while maintaining detailed balance, and Metropolis-Adjusted Langevin Algorithm (Zhang et al., 2022) also introduces the gradient information and updates all dimensions in parallel by factorizing the joint distribution. When second-order information of the target distribution is available, (Zhang et al., 2012; Rhodes & Gutmann, 2022; Sun et al., 2023a) leverage the Gaussian integral trick (Hubbard, 1959) to transform discrete sampling into continuous sampling, enabling the use of numerous well-developed continuous MCMC samplers.

**Stochastic Localization**   Stochastic localization (SL) (Eldan, 2013; 2020; Chen, 2021; Eldan et al., 2022; Eldan & Shamir, 2022) is a recent method for proving properties of measures such as concentration inequalities and measure decompositions. Chen & Eldan (2022) further developed it into a technique for studying mixing time of Markov chains. Additionally, it naturally produces a sampling framework which has attracted the research interest of many data scientists (El Alaoui et al., 2022; Montanari, 2023; Grenioux et al., 2024; Anari et al., 2024).

The main challenge of sampling via SL lies on Bayes estimation of posterior expectation. For discrete targets, El Alaoui et al. (2022) first use SL sampling from the Sherrington-Kirkpatrick (SK) model, where posterior estimation is obtained by an approximate message passing (AMP) algorithm and a process for minimizing the Thouless-Anderson-Palmer (TAP) free energy via natural gradient descent (NGD). Furthermore, they extended this approach to mean-field models (Alaoui et al., 2023a), spiked models (Montanari & Wu, 2023) and spherical spin glasses (Huang et al., 2024). For a certain class of non log-concave continuous distributions, SLIPS (Grenioux et al., 2024) samples from $q_t$ via Metropolis-Adjusted Langevin Algorithm (Roberts & Tweedie, 1996) to estimate posterior mean. In continuous case, log-concavity of distribution implies easier sampling (Dwivedi et al., 2019). SLIPS proves duality of log-concavity to show its efficiency. However, for binary distributions, quadratic term will collapse into constant ($x^2 = 1$, $x \in \{-1, 1\}^n$) or linear term ($x^2 = x$, $x \in \{0, 1\}^n$), which poses difficulties for theoretical analysis.

## B. Background

In this section we will outline some background of stochastic dynamics of algorithms and our theoretical techniques.

### B.1. Fundamental Definitions of Stochastic Localization

A stochastic localization process $(\nu_t)_t$ is defined as

$$\frac{d\nu_t}{d\nu}(x) = F_t(x),$$

where the functions $F_t$ solve following stochastic differential equations:

$$F_0(x) = 1, dF_t(x) = F_t(x)\langle x - m(\nu_t), C_t dB_t\rangle, \forall x, \tag{22}$$

where $m(\nu) := \int x\nu(dx)$, $(C_t)_t$ is an adapted process which takes values in the space of $n \times n$ positive-definite matrices, $(B_t)_{t\geq 0}$ is a standard Brownian motion on $\mathbb{R}^n$. We can also use linear-tilt form describing stochastic localization process

$$\frac{\nu_t(dx)}{\nu(dx)} = e^{Z_t - \frac{1}{2}\langle \Sigma_t x, x\rangle + \langle y_t, x\rangle},$$

where $\Sigma_t = \int_0^t C_s^2 ds$, $y_t = \int_0^t \left(C_s dB_s + C_s^2 m(\nu_s) ds\right)$ and $Z_t$ is a normalizing constant.

A key property is that for any measurable set $A \subset \Omega$, stochastic localization process $\nu_t(A)$ almost surely converges to either 0 or 1 as $t \to \infty$ and the concentrated point is distributed according to the law $\nu$.

**Lemma B.1** (Proposition 10 of (Eldan & Shamir, 2022)). *Let $a_t = m(\nu_t)$, the process $a_t$ almost surely converges to a point in $\Omega$, and $a_\infty := \lim_{t \to \infty} a_t$ is distributed according to the law $\nu$. Moreover, the measure $\nu_t$ almost-surely weakly converges to a Dirac measure at $a_\infty$.*

Lemma B.1 suggest samples $a_\infty \sim \nu$ can be obtained by a stochastic localization process.

### B.2. Generator and Dirichlet Form of Glauber Dynamics and Metropolis-Hastings Algorithms

Consider $(E, d)$ is a polish space equipped with the Borel field $\mathcal{B}$, and the Gibbs measures $\nu$ on spin space $E^T$.

**Glauber Dynamics**   is a Markov process of pure jumps. If the configuration at present is $x$, then at each site $i$, it will change to $x^i$ according to the conditional distribution $\nu(x_i \mid x_{\sim i})$. The generator of Glauber dynamics is

$$\mathcal{L}_{\text{GD}} f(x) = \sum_{i \in T} \left( \mathbb{E}_\nu[f(X) \mid X_{\sim i} = x_{\sim i}] - f(x) \right).$$

And the associated Dirichlet form of Glauber dynamics is

$$\mathcal{E}_{\text{GD}}(f, f) = \mathbb{E}_\nu \sum_{i \in T} (\mathbb{E}_\nu[f(X) \mid X_{\sim i}] - f(X))^2. \tag{23}$$

**Metropolis-Hastings Algorithms**   is a reversible Markov chain with transition function:

$$P(y \mid x) = \Psi(y \mid x) \min\{1, \frac{\nu(y)\Psi(x \mid y)}{\nu(x)\Psi(y \mid x)}\}, \text{ if } x \neq y,$$

and

$$P(y \mid x) = 1 - \sum_{x \neq y} \Psi(y \mid x) \min\{1, \frac{\nu(y)\Psi(x \mid y)}{\nu(x)\Psi(y \mid x)}\}, \text{ if } x = y,$$

where $\Psi$ is the proposal distribution. The generator of Metropolis-Hastings algorithms is

$$\mathcal{L}_{\text{MH}} f(x) = \int_{E^T} P(y \mid x)(f(y) - f(x)).$$

And the associated Dirichlet form of Metropolis-Hastings algorithms is

$$\mathcal{E}_{\text{MH}}(f, f) = \frac{1}{2} \int_{E^T \times E^T} \nu(\mathrm{d}x) P(\mathrm{d}y \mid x) \left( f(x) - f(y) \right)^2. \tag{24}$$

### B.3. Dobrushin Interdependence Matrix

Interdependence matrix serves as a fundamental tool in our analysis, providing a rigorous way to characterize dependency relationships among random variables. This concept, deeply rooted in probability theory and statistical physics, allows us to quantify the strength of interactions between variables. We now present its formal definition.

$L^p$-**Wasserstein Distance**   Let $(\Omega, d)$ be a Polish space equipped with the Borel field $\mathcal{B}$. For $p \geq 1$, we denote by $\mathcal{P}d, p(\Omega)$ the space of probability measures on $(\Omega, \mathcal{B})$ with finite $p$-th moment. For any $\nu_1, \nu_2 \in \mathcal{P}d, p(\Omega)$, the $L^p$-*Wasserstein distance* is defined as

$$\mathcal{W}_{p,d}(\nu_1, \nu_2) := \inf_\pi \left( \int \int_{\Omega \times \Omega} d(x, y)^p \pi(dx, dy) \right)^{1/p}, \tag{25}$$

where the infimum is taken over all probability measures $\pi$ on $\Omega \times \Omega$ with marginals $\nu_1$ and $\nu_2$ respectively. In the special case where $d(x, y) = \mathbf{1}_{x \neq y}$, we have

$$\mathcal{W}_{1,d}(\nu_1, \nu_2) = \sup_{A \in \mathcal{B}} |\nu_1(A) - \nu_2(A)| = \frac{1}{2} \|\nu_1 - \nu_2\|_{TV}.$$

$d$-**Dobrushin Interdependence Matrix**  For $i \in [N]$, let $\nu_i(\cdot|x)$ denote the conditional distribution of $x_i$ given $x_{\sim i}$. The *d-Dobrushin interdependence matrix* $C := (c_{ij})_{i,j \in [N]}$ is defined by

$$c_{ij} := \sup_{x=y \text{ off } j} \frac{\mathcal{W}_{1,d}(\nu_i(\cdot \mid x), \nu_i(\cdot \mid y))}{d(x_j, y_j)}, \quad i, j \in [N], \tag{26}$$

where each entry $c_{ij}$ quantifies the influence of site $j$ on site $i$. The classical Dobrushin uniqueness condition (Dobruschin, 1968; Dobrushin, 1970), $\sup_i \sum_j c_{ij} < 1$, provides a sufficient condition for the uniqueness of Gibbs measures in spin glasses. For a more general generator on $\{-1,1\}^N$ given by

$$\mathcal{L}f(x) = \sum_{S \subset [N]} \int_{-1,1^S} J_S(x, dz_S)(f(x^S) - f(x)), \tag{27}$$

where $J_S$ is a bounded nonnegative kernel representing the local jump rate, the Dobrushin interdependence matrix takes the form

$$c_{ij} := \sum_{i \in S} c_S(j). \tag{28}$$

Here, $c_S(j) \geq 0$ is the optimal constant satisfying

$$\sup_{x=y \text{off} j} \frac{1}{d(x_j, y_j)} | \int_{E^S} g(z_S)(J_S(x, dz_S) - J_S(y, dz_S))|$$

$$\leq c_S(j) \sum_{i \in S} \delta_i(g)$$

for all Lipschitz continuous functions $g$, where $\delta_i(g) := \sup_{x=y \text{ off } i} \frac{|g(y)-g(x)|}{d(y_i, x_i)}$. When $d$ is the trivial metric, we have

$$c_S(j) \leq \frac{1}{2} \sup_{x=y \text{ off } j} \|J_S(x, \cdot) - J_S(y, \cdot)\|_{TV}.$$

## B.4. Poincaré Inequalities for Gibbs Measures under the Dobrushin Uniqueness Condition

In this subsection, we present the mathematical techniques employed in our study. For stochastic dynamics with a given generator, Wu (2006) derives a sharp estimate of the spectral gap (or Poincaré constant) through the spectral radius analysis of the Dobrushin interdependence matrix.

Consider the generator

$$\mathcal{L}f := \sum_{i \in T}[\nu_i(f) - f],$$

where $\nu_i := \nu_i(dx_i|x)$ is the local specification of Gibbs measure $\nu$ on $E^T$, that is, for each $i \in T$ the conditional distribution of $x_i$ knowing $x_T \backslash \{i\}$ coincides with the given $\nu_i(\cdot|x)$. It generates a Glauber dynamics. Then we have following Poincaré inequality for Glauber dynamics.

**Lemma B.2** (Theorem 2.1 of (Wu, 2006)). *Let $r_{sp}(C)$ be the spectral radius of the Dobrushin interdependence matrix $C = (c_{ij})_{i,j \in T}$ defined in* (26) *(which is an eigenvalue of $C$ by the Perron–Frobenius theorem). If $r_{sp}(C) < 1$, then*

$$(1 - r_{sp}(C)) \nu(f,f) \leq \mathbb{E}_\nu \sum_{i \in T} \nu_i(f,f) \quad \forall f \in L^2(E^T, \nu),$$

*where $\nu(f,g)$ denotes the covariance of $f, g$ under $\nu$, and $\nu_i(f,g) = \nu_i(fg) - \nu_i(f)\nu_i(g)$ is the conditional covariance of $f, g$ under $\nu_i$ with $x_{T \backslash \{i\}}$ fixed.*

For more general generator (27)

$$\mathcal{L}f(x) = \sum_{S \subset N} \int_{E^S} J_S(x, dz_S)(f(x^S) - f(x)),$$

Assume that for each $S \subset T$ and for every $j \in T$, there is some finite optimal constant $C_S(j) \geq 0$,

$$\sup_{x=y \text{ off } j} \frac{1}{d(x_j, y_j)} | \int_{E^S} g(z_S) \left(J_S(x, dz_S)\right) - J_S(y, dz_S)| \leq c_S(j) \sum_{i \in S} \delta_i(g)$$

for all Lipschitz continuous function $g$ on $E^T$. And

$$\delta_i(f) := \sup_{x=y \text{ off } i} \frac{|f(y) - f(x)|}{d(y_i, x_i)}.$$

**Lemma B.3** (Theorem 3.1 and Corollary 3.5 of (Wu, 2006)). *Let*

$$\eta := \inf_{x \in E^T} \inf_{i \in T} \sum_{S \ni i} J_S(x, E^S).$$

*For Dobrushin interdependence matrix $C$ defined in (28), assume $r_{\mathrm{sp}}(C) < 1$. Then Markov semigroup $P_t = e^{t\mathcal{L}}$ has a unique invariant measure $\nu$ such that $\int_{E^T} d(x, y)\nu(dy) < +\infty$ for every $x$. Moreover, for each $x \in E^T$,*

$$\mathcal{W}_{1,d}\left(P_t(x, \cdot), \nu\right) \leq e^{-\eta t} \max_j \sum_i (e^{tC})_{ij} \int_{E^T} d(x, y)\nu(dy) \leq e^{-t(\eta - \|C\|_1)} \int_{E^T} d(x, y)\nu(dy).$$

# C. Proofs for Main Theorems

**General Proof Sketch.** Our proof of the Poincaré inequality follows three key steps.

- First, we construct the $d$-Dobrushin interdependence matrix $C$ from the algorithm's transition kernel;

- We then establish upper bounds on the matrix entries $c_{ij}$, which leads to a bound on the spectral radius of $C$.

- Finally, we leverage this spectral bound to establish the Poincaré inequality.

## C.1. Proof for Theorem 3.1

*Proof.* Define the Gaussian integration

$$\Phi(x) = \frac{1}{\sqrt{2\pi}} \int_{-\infty}^x e^{-\frac{y^2}{2}} dy.$$

By definition

$$\begin{aligned}
\mathbb{P}(|Y_t| \geq \zeta) =& \mathbb{P}(|\alpha(t)X + \sigma B_t| \geq \zeta) = \mathbb{P}(|B_t \pm \frac{\alpha(t)}{\sigma}| \geq \frac{\zeta}{\sigma}) \\
=& 1 - \left(\Phi\left(\frac{\zeta}{\sigma\sqrt{t}} \pm \frac{\alpha(t)}{\sigma\sqrt{t}}\right) - \Phi\left(-\frac{\zeta}{\sigma\sqrt{t}} \pm \frac{\alpha(t)}{\sigma\sqrt{t}}\right)\right).
\end{aligned}$$

Obviously,

$$0 \leq \Phi\left(\frac{\zeta}{\sigma\sqrt{t}} \pm \frac{\alpha(t)}{\sigma\sqrt{t}}\right) - \Phi\left(-\frac{\zeta}{\sigma\sqrt{t}} \pm \frac{\alpha(t)}{\sigma\sqrt{t}}\right) \leq \Phi\left(\frac{\zeta}{\sigma\sqrt{t}} - \frac{\alpha(t)}{\sigma\sqrt{t}}\right) \leq \Phi\left(\frac{\zeta}{\sigma} - \frac{\alpha(t)}{\sigma\sqrt{t}}\right).$$

Because $\frac{\alpha(t)}{\sigma\sqrt{t}} \to +\infty$, there is a $T$ large enough, such that for $t \geq T$, we have

$$0 \leq \Phi\left(\frac{\zeta}{\sigma} - \frac{\alpha(t)}{\sigma\sqrt{t}}\right) \leq \varepsilon.$$

Finally, we know that for arbitrary $\zeta > 0$ and $\varepsilon > 0$, when $t \geq T(\alpha, \varepsilon)$

$$\mathbb{P}(|Y_t| \geq \zeta) \geq 1 - \varepsilon.$$

$\square$

## C.2. Proof for Theorem 4.3

*Proof.* The conditional measure $\nu(x_i \mid x_{\sim i})$ is

$$
\begin{aligned}
\nu_{\beta,h}(x_i \mid x_{\sim i}) &= \frac{e^{-\frac{\beta}{2}\langle x, Wx\rangle + \langle x, h\rangle}}{\sum_{x_i = \pm 1} e^{-\frac{\beta}{2}\langle x, Wx\rangle + \langle x, h\rangle}} \\
&= \frac{e^{-\frac{\beta}{2}\sum_{k\neq i} W_{ik}x_i x_k - \frac{\beta}{2}W_{ii} - \frac{\beta}{2}\sum_{k,j\neq i} W_{jk}x_j x_k + h_i x_i + \sum_{k\neq i} h_k x_k}}{\sum_{x_i=\pm 1} e^{-\frac{\beta}{2}\sum_{k\neq i} W_{ik}x_i x_k - \frac{\beta}{2}W_{ii} - \frac{\beta}{2}\sum_{k,j\neq i} W_{jk}x_j x_k + h_i x_i + \sum_{k\neq i} h_k x_k}} \\
&= \frac{e^{-\frac{\beta}{2}\sum_{k\neq i} W_{ik}x_i x_k + h_i x_i}}{e^{-\frac{\beta}{2}\sum_{k\neq i} W_{ik}x_k + h_i} + e^{\frac{\beta}{2}\sum_{k\neq i} W_{ik}x_k - h_i}}.
\end{aligned}
$$

Because

$$
\mathcal{W}_{1,d}(\nu_{\beta,h}(x_i \mid x_{\sim i}), \nu_{\beta,h}(x_i \mid \hat{x}_{\sim i})) = \sup_{A \in \mathcal{B}} |\nu_{\beta,h}(A \mid x_{\sim i}) - \nu_{\beta,h}(A \mid \hat{x}_{\sim i})|,
$$

where $\mathcal{B} = \{\emptyset, \{+1, -1\}, \{+1\}, \{-1\}\}$. For $A = \{+1\}$,

$$
\begin{aligned}
&|\nu_{\beta,h}(x_i = +1 \mid x_{\sim i}) - \nu_{\beta,h}(x_i = +1 \mid \hat{x}_{\sim i})| \\
&= \left| \frac{e^{-\frac{\beta}{2}\sum_{k\neq i,j} W_{ik}x_k + h_i - \frac{\beta}{2}W_{ij}x_j}}{2\cosh\left(-\frac{\beta}{2}\sum_{k\neq i,j} W_{ik}x_k + h_i - \frac{\beta}{2}W_{ij}x_j\right)} - \frac{e^{-\frac{\beta}{2}\sum_{k\neq i,j} W_{ik}x_k + h_i - \frac{\beta}{2}W_{ij}\hat{x}_j}}{2\cosh\left(-\frac{\beta}{2}\sum_{k\neq i,j} W_{ik}x_k + h_i - \frac{\beta}{2}W_{ij}\hat{x}_j\right)} \right|,
\end{aligned}
$$

and for $A = \{-1\}$,

$$
\begin{aligned}
&|\nu_{\beta,h}(x_i = -1 \mid x_{\sim i}) - \nu_{\beta,h}(x_i = -1 \mid \hat{x}_{\sim i})| \\
&= \left| \frac{e^{\frac{\beta}{2}\sum_{k\neq i,j} W_{ik}x_k - h_i + \frac{\beta}{2}W_{ij}x_j}}{2\cosh\left(-\frac{\beta}{2}\sum_{k\neq i,j} W_{ik}x_k + h_i - \frac{\beta}{2}W_{ij}x_j\right)} - \frac{e^{\frac{\beta}{2}\sum_{k\neq i,j} W_{ik}x_k - h_i + \frac{\beta}{2}W_{ij}\hat{x}_j}}{2\cosh\left(-\frac{\beta}{2}\sum_{k\neq i,j} W_{ik}x_k + h_i - \frac{\beta}{2}W_{ij}\hat{x}_j\right)} \right|.
\end{aligned}
$$

Consider function

$$
g_1(x) = \frac{e^x}{2\cosh(x)}, g_2(x) = \frac{e^{-x}}{2\cosh(x)},
$$

we know that

$$
g_1'(x) = -g_2'(x) = \frac{2}{(e^x + e^{-x})^2}.
$$

Taking $h_0 \geq 0$ such that

$$
\beta \sup_{i \in T} \sum_{k\neq i} |W_{ik}| \leq h_0,
$$

then

$$
\left| \frac{\beta}{2}\sum_{k\neq i,j} W_{ik}x_k + \frac{\beta}{2}W_{ij}x_j \right| \leq \frac{\beta}{2}\sup_{i\in T}\sum_{k\neq i}|W_{ik}| \leq \frac{h_0}{2}.
$$

Hence, for $h = 2h_0$

$$
\frac{3h_0}{2} = h - \frac{h_0}{2} \leq -\frac{\beta}{2}\sum_{k\neq i,j} W_{ik}x_k - \frac{\beta}{2}W_{ij}x_j + h \leq \frac{h_0}{2} + h,
$$

and for $h = -2h_0$

$$h - \frac{h_0}{2} \leq -\frac{\beta}{2} \sum_{k \neq i,j} W_{ik}x_k - \frac{\beta}{2}W_{ij}x_j + h \leq \frac{h_0}{2} + h = -\frac{3h_0}{2}.$$

Putting everything together we get for $|h| = 2h_0$

$$\left| -\frac{\beta}{2} \sum_{k \neq i,j} W_{ik}x_k - \frac{\beta}{2}W_{ij}x_j + h \right| \geq \frac{3h_0}{2}.$$

and

$$g_1' \left( -\frac{\beta}{2} \sum_{k \neq i,j} W_{ik}x_k - \frac{\beta}{2}W_{ij}x_j + h \right) \leq \frac{2}{e^{\frac{3h_0}{2}} + e^{-\frac{3h_0}{2}}} = \frac{2}{e^{\frac{3|h|}{4}} + e^{-\frac{3|h|}{4}}}.$$

Hence, for

$$2\beta \sup_{i \in T} \sum_{k \neq i} |W_{ik}| \leq |h|,$$

we get

$$\mathcal{W}_{1,d}(\nu_{\beta,h}(x_i \mid x_{\sim i}), \nu_{\beta,h}(x_i \mid \hat{x}_{\sim i})) = \sup_{A \in \mathcal{B}} |\nu_{\beta,h}(A \mid x_{\sim i}) - \nu_{\beta,h}(A \mid \hat{x}_{\sim i})|$$

$$\leq \frac{2}{e^{\frac{3|h|}{4}} + e^{-\frac{3|h|}{4}}} \left| \frac{\beta}{2}W_{ij}(x_j - \hat{x}_j) \right| \leq \frac{2\beta|W_{ij}|}{e^{\frac{3|h|}{4}} + e^{-\frac{3|h|}{4}}},$$

which implies

$$c_{ij} := \sup_{x=y \text{ off } j} \frac{\mathcal{W}_{1,d}(\nu_i(\cdot \mid x), \nu_i(\cdot \mid y))}{d(x_j, y_j)} \leq \frac{2\beta|W_{ij}|}{e^{\frac{3|h|}{4}} + e^{-\frac{3|h|}{4}}}, i, j \in [N].$$

Hence,

$$\|C\|_\infty = \sup_i \sum_{j \in [N]} c_{ij} \leq \sup_i \sum_{j \in [N]} \frac{2\beta|W_{ij}|}{e^{\frac{3|h|}{4}} + e^{-\frac{3|h|}{4}}} \leq \frac{|h|}{e^{\frac{3|h|}{4}} + e^{-\frac{3|h|}{4}}}.$$

Therefore

$$r_{\mathrm{sp}}(C) \leq \|C\|_\infty \leq \frac{|h|}{e^{\frac{3|h|}{4}} + e^{-\frac{3|h|}{4}}}.$$

Combine with Lemma B.2, we have

$$\mathrm{Var}_{\nu_{\beta,h}}(f) \leq \frac{1}{1 - \frac{|h|}{e^{\frac{3|h|}{4}} + e^{-\frac{3|h|}{4}}}} \mathcal{E}_{GD}(f, f).$$

$\square$

### C.3. Proof of Corollary 4.4

*Proof.* The following lemma gives a Chernoff-type bound for Markov chains.

**Lemma C.1** (Theorem 3.6 of (Dubhashi & Panconesi, 2009)). *Let $X_1, X_2, \ldots, X_n$ be a sequence generated by a irreducible and reversible Markov chain on a finite set with invariant distribution $\pi$ and spectral gap $\gamma$. Then, for any initial distribution $q$; any positive integer $n$ and all $\epsilon > 0$,*

$$P_q\left[\left|\sum_{i=1}^n f(X_i) - n\pi(f)\right| \geq \varepsilon\right] \leq C_{\gamma,n,q} \exp\left(-\frac{\varepsilon^2\gamma}{cn}\right),$$

*where $c$ is an absolute constant and $C_{\gamma,n,q}$ is a rational function. Lezaud (1998); Gillman (1998) also prove the same result.*

It is well-known that Glauber dynamics is reversible with respect to Gibbs measure $\nu_{\beta,h}$ (Bertoin et al., 1999). And we consider sampling from a finite set $\{-1, 1\}^N$, thus Glauber dynamics is irreducible. From lemma C.1, we have

$$P_q\left[\left|\frac{1}{n}\sum_{i=1}^n X_i - \mathbb{E}_{\nu_{\beta,h}}[X]\right| \geq \varepsilon\right] \leq C_{\gamma_{\text{gap}},n,q} e^{-\frac{n\varepsilon^2 \gamma_{\text{gap}}}{c}},$$

where $\gamma_{\text{gap}}$ is spectral gap of Glauber dynamics corresponding to $\nu_{\beta,h}$. Combining with Theorem 4.3, we have $\gamma_{\text{gap}} = 1 - \frac{|h|}{e^{\frac{3|h|}{4}} + e^{-\frac{3|h|}{4}}}$.

$\square$

## C.4. Proof of Theorem 4.5

*Proof.* For $i \in T$, $x \to x^i$ with probability

$$P(x^i \mid x) = \min\{1, \frac{\nu_{\beta,h}(x^i)}{\nu_{\beta,h}(x)}\} = \min\{1, e^{2\beta x_i \sum_{j\neq i} W_{ij}x_j - 2h_i x_i}\},$$

and keep invariant with probability

$$1 - P(x^i \mid x) = 1 - \min\{1, e^{2\beta x_i \sum_{j\neq i} W_{ij}x_j - 2h_i x_i}\},$$

Then

$$\mathcal{L}f(x) = \int_{E^T}(f(y) - f(x))p(y \mid x) = \sum_{i=1}^T \int_{E^i}(f(x^{y_i}) - f(x))P(x^{y_i} \mid x)$$

$$= \sum_{i=1}^T (f(x^i) - f(x))\min\{1, e^{2\beta x_i \sum_{j\neq i} W_{ij}x_j - 2h_i x_i}\},$$

Define the local kernel as follows

$$\nu_i(x, x^i) = \min\{1, e^{2\beta x_i \sum_{j\neq i} W_{ij}x_j - 2h_i x_i}\},$$

Obviously, $\nu_i(x, E^i) = 1$ for any $x$, then $\eta = 1$.

$$c_i(j) = \sup_{x=\hat{x} \text{ off } j} |\nu_i(x, \cdot) - \nu_i(\hat{x}, \cdot)|_{TV} = \sup_{x=\hat{x} \text{ off } j} \sup_{\mathcal{B}} |\nu_i(x, B) - \nu_i(\hat{x}, B)|$$

$$= \sup_{x=\hat{x} \text{ off } j} \sup_{\mathcal{B}_i} |\nu_i(x_i, A) - \nu_i(\hat{x}_i, A)|$$

$$= \sup_{x=\hat{x} \text{ off } j} \max\{\left|\min\{1, e^{2\beta x_i \sum_{k\neq i,j} W_{ik}x_k + 2\beta x_i W_{ij}x_j - 2h_i x_i}\} - \min\{1, e^{2\beta x_i \sum_{k\neq i,j} W_{ik}x_k - 2\beta x_i W_{ij}x_j - 2h_i x_i}\}\right|,$$

$$\left|1 - \min\{1, e^{2\beta x_i \sum_{k\neq i,j} W_{ik}x_k + 2\beta x_i W_{ij}x_j - 2h_i x_i}\} - 1 + \min\{1, e^{2\beta x_i \sum_{k\neq i,j} W_{ik}x_k - 2\beta x_i W_{ij}x_j - 2h_i x_i}\}\right|\}$$

$$= \sup_{x=\hat{x} \text{ off } j} \left|\min\{1, e^{2\beta x_i \sum_{k\neq i,j} W_{ik}x_k + 2\beta x_i W_{ij}x_j - 2h_i x_i}\} - \min\{1, e^{2\beta x_i \sum_{k\neq i,j} W_{ik}x_k - 2\beta x_i W_{ij}x_j - 2h_i x_i}\}\right|.$$

Because

$$\left|2\beta \sum_{k\neq i,j} W_{ik}x_k + 2\beta W_{ij}x_j\right| \leq 2\beta \sup_{i\in T}\sum_{k\neq i}|W_{ik}| \leq 2h_0.$$

Hence, for $h = 2h_0$

$$-3h \leq 2\beta \sum_{k\neq i,j} W_{ik}x_k + 2\beta W_{ij}x_j - 2h \leq -h$$

$$-3h \leq 2\beta \sum_{k\neq i,j} W_{ik}x_k - 2\beta W_{ij}x_j - 2h \leq -h,$$

and for $h = -2h_0$

$$2h_0 \leq 2\beta \sum_{k \neq i,j} W_{ik}x_k + 2\beta W_{ij}x_j - 2h \leq 6h_0$$

$$2h_0 \leq 2\beta \sum_{k \neq i,j} W_{ik}x_k - 2\beta W_{ij}x_j - 2h \leq 6h_0.$$

Putting everything together we get for $|h| = 2h_0$

$$\left| 2\beta \sum_{k \neq i,j} W_{ik}x_k \pm 2\beta W_{ij}x_j - 2h \right| \geq 2h_0.$$

Hence,

$$c_i(j) = \sup_{x = \hat{x} \text{ off } j} \left| \min\{1, e^{2\beta x_i \sum_{k \neq i,j} W_{ik}x_k + 2\beta x_i W_{ij}x_j - 2h_i x_i}\} - \min\{1, e^{2\beta x_i \sum_{k \neq i,j} W_{ik}x_k - 2\beta x_i W_{ij}x_j - 2h_i x_i}\} \right|$$

$$\leq e^{-2h_0} 4\beta |W_{ij}| = e^{-|h|} 4\beta |W_{ij}|.$$

Finally,

$$\|C\|_\infty = \sup_i \sum_{j \in [N]} c_i(j) \leq 2|h|e^{-|h|},$$

and by Lemma B.2, we can get

$$\mathrm{Var}_{\nu_{\beta,h}}(f) \leq \frac{1}{1 - 2|h|e^{-|h|}} \mathcal{E}_{MH}(f, f).$$

$\square$

## C.5. Proof of Theorem 4.6

*Proof.* The transition matrix of Metropolis chain is

$$P(y \mid x) = \Psi(x, y) \min\{1, \frac{\nu_{\beta,h}(y)\Psi(x \mid y)}{\nu_{\beta,h}(x)\Psi(y \mid x)}\}, \text{ if } x \neq y,$$

and

$$P(y \mid x) = 1 - \sum_{x \neq y} \Psi(x, y) \min\{1, \frac{\pi(y)\Psi(x \mid y)}{\pi(x)\Psi(y \mid x)}\}, \text{ if } x = y.$$

Particularly, on the configuration $\{+1, -1\}^N$, the dynamic flips one site each time, that is for $|x - y| \geq 2$

$$P(y \mid x) = 0,$$

and for $i \in [N]$, $x \to x^i$ with probability

$$P(x^i \mid x) = \Psi(x^i \mid x) \min\{1, \frac{\nu_{\beta,h}(y)\Psi(x \mid x^i)}{\nu_{\beta,h}(x)\Psi(x^i \mid x)}\},$$

and keep invariant with probability

$$1 - P(x^i \mid x) = 1 - \Psi(x^i \mid x) \min\{1, \frac{\nu_{\beta,h}(x^i)\Psi(x \mid x^i)}{\nu_{\beta,h}(x)\Psi(x^i \mid x)}\},$$

Then

$$\mathcal{L}f(x) = \int_{E^T} (f(y) - f(x))p(y \mid x) = \sum_{i=1}^{T} \int_{E^i} (f(x^{y_i}) - f(x))P(x^{y_i} \mid x)$$

$$= \sum_{i=1}^{T} (f(x^i) - f(x))\Psi(x^i \mid x) \min\{1, \frac{\pi(x^i)\Psi(x \mid x^i)}{\pi(x)\Psi(x^i \mid x)}\},$$

Define the local kernel as follows

$$\nu_i(x, x^i) = \Psi(x^i \mid x) \min\{1, \frac{\nu_{\beta,h}(x^i)\Psi(x \mid x^i)}{\nu_{\beta,h}(x)\Psi(x^i \mid x)}\}.$$

Obviously, $\nu_i(x, E^i) = 1$ for any $x$, then $\eta = 1$.

For $\mathcal{B} = \{+1, -1\}^N$, $i \neq j$

$$c_i(j) = \sup_{x=\hat{x} \text{ off } j} |\nu_i(x, \cdot) - \nu_i(\hat{x}, \cdot)|_{TV} = \sup_{x=\hat{x} \text{ off } j} \sup_{\mathcal{B}} |\nu_i(x, B) - \nu_i(\hat{x}, B)|$$

$$= \sup_{x=\hat{x} \text{ off } j} \sup_{\mathcal{B}_i} |\nu_i(x_{\sim i}, A) - \nu_i(\hat{x}_{\sim i}, A)|$$

$$= \sup_{x=\hat{x} \text{ off } j} \max\{|\nu_i(x_{\sim i}, +1) - \nu_i(\hat{x}_{\sim i}, +1)|, |\nu_i(x_{\sim i}, -1) - \nu_i(\hat{x}_{\sim i}, -1)|\}$$

$$= \sup_{x=\hat{x} \text{ off } j} \max\{\left|\Psi(x^i \mid x) \min\{1, \frac{\nu_{\beta,h}(x^i)\Psi(x \mid x^i)}{\nu_{\beta,h}(x)\Psi(x^i \mid x)}\} - \Psi(\hat{x}^i \mid \hat{x}) \min\{1, \frac{\nu_{\beta,h}(\hat{x}^i)\Psi(\hat{x} \mid \hat{x}^i)}{\nu_{\beta,h}(\hat{x})\Psi(\hat{x}^i \mid \hat{x})}\}\right|,$$

$$\left|1 - \Psi(x^i \mid x) \min\{1, \frac{\nu_{\beta,h}(x^i)\Psi(x \mid x^i)}{\nu_{\beta,h}(x)\Psi(x^i \mid x)}\} - 1 + \Psi(\hat{x}^i \mid \hat{x}) \min\{1, \frac{\nu_{\beta,h}(\hat{x}^i)\Psi(\hat{x} \mid \hat{x}^i)}{\nu_{\beta,h}(\hat{x})\Psi(\hat{x}^i \mid \hat{x})}\}\right|\}$$

$$= \sup_{x=\hat{x} \text{ off } j} \left|\Psi(x^i \mid x) \min\{1, \frac{\nu_{\beta,h}(x^i)\Psi(x \mid x^i)}{\nu_{\beta,h}(x)\Psi(x^i \mid x)}\} - \Psi(\hat{x}^i \mid \hat{x}) \min\{1, \frac{\nu_{\beta,h}(\hat{x}^i)\Psi(\hat{x} \mid \hat{x}^i)}{\nu_{\beta,h}(\hat{x})\Psi(\hat{x}^i \mid \hat{x})}\}\right| \leq C_{Lip}(\beta, h)|\beta W_{ij}|.$$

Hence,

$$\|C\|_\infty = \sup_i \sum_{j \in [N]} c_i(j) \leq C_{Lip}(\beta, h)|h|,$$

which implies the Poincaré inequality

$$\text{Var}_{\nu_{\beta,h}}(f) \leq \frac{1}{1 - C_{Lip}(\beta, h)|h|} \mathcal{E}_{MH}(f, f).$$

$\square$

## C.6. Proof of Theorem 4.8

*Proof.* For Gibbs measure

$$\nu_{\beta,h} = \frac{1}{Z_{\beta,h}} e^{-\frac{\beta}{2}\langle x, Wx \rangle + \langle x, h \rangle},$$

we consider the following with kernel

$$P(x^i|x) = \Psi(x^i \mid x) \min\{1, \frac{\nu_{\beta,h}(x^i)\Psi(x \mid x^i)}{\nu_{\beta,h}(x)\Psi(x^i \mid x)}\},$$

where

$$\Psi(x^i \mid x) = \frac{1}{Z_{\beta,h}(x)} e^{-\frac{\beta}{2}\langle x, W(x^i-x) \rangle + \frac{1}{2}\langle x^i-x, h \rangle} = \frac{e^{\beta x_i \sum_{j \neq i} W_{ij}x_j - hx_i}}{Z_{\beta,h}(x)},$$

and

$$Z_{\beta,h}(x) = 1 + e^{\beta x_i \sum_{j \neq i} W_{ij} x_j - h x_i},$$

then

$$\Psi(x^i \mid x) = \frac{1}{Z_{\beta,h}(x)} e^{-\frac{\beta}{2}\langle x, W(x^i - x)\rangle + \frac{1}{2}\langle x^i - x, h\rangle} = \frac{e^{\beta x_i \sum_{j \neq i} W_{ij} x_j - h x_i}}{1 + e^{\beta x_i \sum_{j \neq i} W_{ij} x_j - h x_i}}$$

$$= \frac{e^{\frac{1}{2}\beta x_i \sum_{j \neq i} W_{ij} x_j - \frac{1}{2} h x_i}}{e^{-\frac{1}{2}\beta x_i \sum_{j \neq i} W_{ij} x_j + \frac{1}{2} h x_i} + e^{\frac{1}{2}\beta x_i \sum_{j \neq i} W_{ij} x_j - \frac{1}{2} h x_i}},$$

and

$$\Psi(x \mid x^i) = \frac{e^{-\frac{1}{2}\beta x_i \sum_{j \neq i} W_{ij} x_j + \frac{1}{2} h x_i}}{e^{-\frac{1}{2}\beta x_i \sum_{j \neq i} W_{ij} x_j + \frac{1}{2} h x_i} + e^{\frac{1}{2}\beta x_i \sum_{j \neq i} W_{ij} x_j - \frac{1}{2} h x_i}}.$$

Hence,

$$\frac{\Psi(x \mid x^i)}{\Psi(x^i \mid x)} = \frac{e^{-\frac{1}{2}\beta x_i \sum_{j \neq i} W_{ij} x_j + \frac{1}{2} h x_i}}{e^{\frac{1}{2}\beta x_i \sum_{j \neq i} W_{ij} x_j - \frac{1}{2} h x_i}} = e^{-\beta x_i \sum_{j \neq i} W_{ij} x_j + h x_i}.$$

Furthermore,

$$\Psi(\hat{x}^i \mid \hat{x}) = \frac{e^{\frac{1}{2}\beta x_i \sum_{k \neq i,j} W_{ik} x_k - \frac{1}{2}\beta x_i W_{ij} x_j - \frac{1}{2} h x_i}}{e^{-\frac{1}{2}\beta x_i \sum_{k \neq i,j} W_{ik} x_k + \frac{1}{2}\beta x_i W_{ij} x_j + \frac{1}{2} h x_i} + e^{\frac{1}{2}\beta x_i \sum_{k \neq i,j} W_{ik} x_k - \frac{1}{2}\beta x_i W_{ij} x_j - \frac{1}{2} h x_i}}$$

$$\Psi(\hat{x} \mid \hat{x}^i) = \frac{e^{-\frac{1}{2}\beta x_i \sum_{k \neq i,j} W_{ik} x_k + \frac{1}{2}\beta x_i W_{ij} x_j + \frac{1}{2} h x_i}}{e^{-\frac{1}{2}\beta x_i \sum_{k \neq i,j} W_{ik} x_k + \frac{1}{2}\beta x_i W_{ij} x_j + \frac{1}{2} h x_i} + e^{\frac{1}{2}\beta x_i \sum_{k \neq i,j} W_{ik} x_k - \frac{1}{2}\beta x_i W_{ij} x_j - \frac{1}{2} h x_i}}.$$

and

$$\frac{\Psi(\hat{x} \mid \hat{x}^i)}{\Psi(\hat{x}^i \mid \hat{x})} = e^{-\beta x_i \sum_{k \neq i,j} W_{ik} x_k + \beta x_i W_{ij} x_j + h x_i}.$$

Besides,

$$\frac{\nu_{\beta,h}(x^i)}{\nu_{\beta,h}(x)} = e^{2\beta x_i \sum_{k \neq i,j} W_{ik} x_k + 2\beta x_i W_{ij} x_j - 2 h x_i}, \quad \frac{\nu_{\beta,h}(\hat{x}^i)}{\nu_{\beta,h}(\hat{x})} = e^{2\beta x_i \sum_{k \neq i,j} W_{ik} x_k - 2\beta x_i W_{ij} x_j - 2 h x_i},$$

and

$$\frac{\nu_{\beta,h}(x^i)}{\nu_{\beta,h}(x)} \frac{\Psi(x \mid x^i)}{\Psi(x^i \mid x)} = e^{-\beta x_i \sum_{k \neq i,j} W_{ik} x_k - \beta x_i W_{ij} x_j + h x_i} e^{2\beta x_i \sum_{k \neq i,j} W_{ik} x_k + 2\beta x_i W_{ij} x_j - 2 h x_i}$$

$$= e^{\beta x_i \sum_{k \neq i,j} W_{ik} x_k + \beta x_i W_{ij} x_j - h x_i}$$

$$\frac{\nu_{\beta,h}(\hat{x}^i)}{\nu_{\beta,h}(\hat{x})} \frac{\Psi(\hat{x} \mid \hat{x}^i)}{\Psi(\hat{x}^i \mid \hat{x})} = e^{\beta x_i \sum_{k \neq i,j} W_{ik} x_k - \beta x_i W_{ij} x_j - h x_i}.$$

Consider the function

$$f(x, y) = \frac{\min\{1, e^{x+y}\}}{1 + e^{x+y}} = \frac{1}{2} - \frac{|1 - e^{x+y}|}{2(1 + e^{x+y})},$$

then for $y_1, y_2 \in (-\infty, -x) \cup (-x, +\infty)$

$$|f(x, y_1) - f(x, y_2)| \leq \frac{1}{(e^{\frac{x+y}{2}} + e^{-\frac{x+y}{2}})^2} |y_2 - y_1|,$$

for $y_1 \leq -x \leq y_2$

$$|f(x, y_1) - f(x, y_2)| \leq |f(x, y_1) - f(x, -x)| \leq \frac{1}{(e^{\frac{x+y}{2}} + e^{-\frac{x+y}{2}})^2} |-x - y_1|$$

$$\leq \frac{1}{(e^{\frac{x+y}{2}} + e^{-\frac{x+y}{2}})^2} |y_2 - y_1|.$$

Denote that

$$x = \beta x_i \sum_{k \neq i,j} W_{ik} x_k - h x_i$$

$$y = \beta x_i W_{ij} x_j.$$

Recall that for $h = 2h_0$

$$\frac{h_0}{2} = \frac{h}{2} - \frac{h_0}{2} \leq -\frac{\beta}{2} \sum_{k \neq i,j} W_{ik} x_k - \frac{\beta}{2} W_{ij} x_j + \frac{h}{2} \leq \frac{h_0}{2} + \frac{h}{2},$$

and for $h = -2h_0$

$$\frac{h}{2} - \frac{h_0}{2} \leq -\frac{\beta}{2} \sum_{k \neq i,j} W_{ik} x_k - \frac{\beta}{2} W_{ij} x_j + \frac{h}{2} \leq \frac{h_0}{2} + \frac{h}{2} = -\frac{h_0}{2}.$$

Putting everything together we get for $|h| = 2h_0$

$$\left| -\frac{\beta}{2} \sum_{k \neq i,j} W_{ik} x_k - \frac{\beta}{2} W_{ij} x_j + \frac{h}{2} \right| \geq \frac{h_0}{2}.$$

Hence,

$$\left| \Psi(x^i \mid x) \min\{1, \frac{\nu_{\beta,h}(x^i)\Psi(x \mid x^i)}{\nu_{\beta,h}(x)\Psi(x^i \mid x)}\} - \Psi(\hat{x}^i \mid \hat{x}) \min\{1, \frac{\nu_{\beta,h}(\hat{x}^i)\Psi(\hat{x} \mid \hat{x}^i)}{\nu_{\beta,h}(\hat{x})\Psi(\hat{x}^i \mid \hat{x})}\} \right|$$

$$= \left| \frac{e^{\frac{1}{2}(x+y)}}{e^{\frac{1}{2}(x+y)} + e^{-\frac{1}{2}(x+y)}} \min\{1, e^{x+y}\} - \frac{e^{\frac{1}{2}(x-y)}}{e^{\frac{1}{2}(x-y)} + e^{-\frac{1}{2}(x-y)}} \min\{1, e^{x-y}\} \right|$$

$$= \left| \frac{1}{1 + e^{-(x+y)}} \min\{1, e^{x+y}\} - \frac{1}{1 + e^{-(x-y)}} \min\{1, e^{x-y}\} \right|$$

$$\leq \frac{2}{(e^{\frac{x+y}{2}} + e^{-\frac{x+y}{2}})^2} |y|$$

$$\leq \frac{2\beta|W_{ij}|}{(e^{\frac{\beta x_i \sum_{k \neq i,j} W_{ik} x_k - h x_i + \beta x_i W_{ij} x_j}{2}} + e^{\frac{-\beta x_i \sum_{k \neq i,j} W_{ik} x_k + h x_i - \beta x_i W_{ij} x_j}{2}})^2}$$

$$\leq \frac{2\beta|W_{ij}|}{(e^{\frac{|h|}{4}} + e^{-\frac{|h|}{4}})^2}$$

Finally, we get the estimate

$$c_i(j) \leq \frac{2\beta|W_{ij}|}{(e^{\frac{|h|}{4}} + e^{-\frac{|h|}{4}})^2}.$$

Using Lemma B.2, we have

$$\mathrm{Var}_{\nu_{\beta,h}}(f) \leq \frac{1}{1 - \frac{|h|}{(e^{\frac{|h|}{4}} + e^{-\frac{|h|}{4}})^2}} \mathcal{E}_{MH}(f, f).$$

$\square$

## C.7. Proof of Theorem 4.9

Define the transition kernel $J_T(x, -x)$ as follows

$$J_T(x, -x) = \prod_{i=1}^{T} \Psi(x^i \mid x) = \prod_{i=1}^{T} \frac{e^{\beta x_i \sum_{j \neq i} W_{ij} x_j - h x_i}}{1 + e^{\beta x_i \sum_{j \neq i} W_{ij} x_j - h x_i}},$$

and

$$\prod_{i=1}^{T} \Psi(\hat{x}^i \mid \hat{x}) = \prod_{i=1}^{T} \frac{e^{\beta \hat{x}_i \sum_{j \neq i} W_{ij} \hat{x}_j - h \hat{x}_i}}{1 + e^{\beta \hat{x}_i \sum_{j \neq i} W_{ij} \hat{x}_j - h \hat{x}_i}},$$

where $x = \hat{x}$ off $j$. Then

$$\prod_{i=1}^{T} \Psi(x^i \mid x) - \prod_{i=1}^{T} \Psi(\hat{x}^i \mid \hat{x})$$

$$= \Psi(x^j \mid x) \prod_{i \neq j}^{T} \Psi(x^i \mid x) - \Psi(\hat{x}^j \mid x) \prod_{i \neq j}^{T} \Psi(\hat{x}^i \mid \hat{x})$$

$$= \Psi(x^j \mid x) \prod_{i \neq j}^{T} \frac{e^{\beta x_i \sum_{k \neq i,j} W_{ik} x_k + \beta x_i W_{ij} x_j - h x_i}}{1 + e^{\beta x_i \sum_{k \neq i,j} W_{ik} x_k + \beta x_i W_{ij} x_j - h x_i}} - \Psi(\hat{x}^j \mid \hat{x}) \prod_{i \neq j}^{T} \frac{e^{\beta x_i \sum_{k \neq i,j} W_{ik} x_k - \beta x_i W_{ij} x_j - h x_i}}{1 + e^{\beta x_i \sum_{k \neq i,j} W_{ik} x_k - \beta x_i W_{ij} x_j - h x_i}}$$

$$:= \prod_{i=1}^{N} f_i(x) - \prod_{i=1}^{N} f_i(\hat{x}),$$

where

$$f_i(x) = \frac{e^{\beta x_i \sum_{k \neq i,j} W_{ik} x_k + \beta x_i W_{ij} x_j - h x_i}}{1 + e^{\beta x_i \sum_{k \neq i,j} W_{ik} x_k + \beta x_i W_{ij} x_j - h x_i}}, i \neq j,$$

and

$$f_j(x) = \Psi(x^j \mid x) = \frac{e^{\beta x_j \sum_{k \neq j} W_{jk} x_k - h x_j}}{1 + e^{\beta x_j \sum_{k \neq j} W_{jk} x_k - h x_j}}, f_j(\hat{x}) = \Psi(\hat{x}^j \mid \hat{x}) = \frac{e^{-\beta x_j \sum_{k \neq j} W_{jk} x_k + h x_j}}{1 + e^{-\beta x_j \sum_{k \neq j} W_{jk} x_k + h x_j}}.$$

Recall that for $h_0 \geq 0$ such that

$$\beta \sup_{i \in T} \sum_{k \neq i} |W_{ik}| \leq h_0,$$

then

$$\left| \beta \sum_{k \neq i,j} W_{ik} x_k \pm \beta W_{ij} x_j \right| \leq \beta \sup_{i \in T} \sum_{k \neq i} |W_{ik}| \leq h_0.$$

Hence, for $h = 2h_0$

$$-3h_0 \leq \beta \sum_{k \neq i,j} W_{ik} x_k \pm \beta W_{ij} x_j - h \leq -h_0,$$

and for $h = -2h_0$

$$h_0 \leq \beta \sum_{k \neq i,j} W_{ik} x_k \pm \beta W_{ij} x_j - h \leq 3h_0.$$

Putting everything together we get for $|h| = 2h_0$

$$\left| \beta \sum_{k \neq i,j} W_{ik}x_k \pm \beta W_{ij}x_j - h \right| \geq h_0.$$

For $|y| \geq h_0$

$$0 \leq \frac{d}{dy}\left( \frac{1}{1+e^y} \right) \leq \frac{1}{(e^{\frac{h_0}{2}} + e^{-\frac{h_0}{2}})^2}.$$

Because for any $i \in [N], 0 < \Psi(x^i \mid x) \leq 1$, then

$$\left| \prod_{i=1}^{N} \Psi(x^i \mid x) - \prod_{i=1}^{N} \Psi(\hat{x}^i \mid \hat{x}) \right| = \left| \Psi(x^1 \mid x) \prod_{i=2}^{N} \Psi(x^i \mid x) - \Psi(\hat{x}^1 \mid \hat{x}) \prod_{i=2}^{N} \Psi(x^i \mid x) \right|$$

$$+ \left| \Psi(\hat{x}^1 \mid \hat{x}) \prod_{i=2}^{N} \Psi(x^i \mid x) - \prod_{i=1}^{N} \Psi(\hat{x}^i \mid \hat{x}) \right|$$

$$= \left| \left( \Psi(x^1 \mid x) - \Psi(\hat{x}^1 \mid \hat{x}) \right) \prod_{i=2}^{N} \Psi(x^i \mid x) + \Psi(\hat{x}^1 \mid \hat{x}) \left( \prod_{i=2}^{N} \Psi(x^i \mid x) - \prod_{i=2}^{N} \Psi(\hat{x}^i \mid \hat{x}) \right) \right|$$

$$\leq \left| \Psi(x^1 \mid x) - \Psi(\hat{x}^1 \mid \hat{x}) \right| + \left| \prod_{i=2}^{N} \Psi(x^i \mid x) - \prod_{i=2}^{N} \Psi(\hat{x}^i \mid \hat{x}) \right|$$

$$\leq \sum_{i=1}^{N} \left| \Psi(x^i \mid x) - \Psi(\hat{x}^i \mid \hat{x}) \right|.$$

For $i \neq j$

$$\left| \Psi(x^i \mid x) - \Psi(\hat{x}^i \mid \hat{x}) \right| \leq \frac{2\beta |W_{ij}|}{(e^{\frac{h_0}{2}} + e^{-\frac{h_0}{2}})^2}.$$

For $i = j$

$$\left| \Psi(x^j \mid x) - \Psi(\hat{x}^j \mid \hat{x}) \right| \leq \frac{1}{(e^{\frac{h_0}{2}} + e^{-\frac{h_0}{2}})^2} \left| 2\beta \sum_{k \neq j} W_{jk}x_k - 2h \right| \leq \frac{2\beta \sum_{k \neq j} |W_{jk}| + 2|h|}{(e^{\frac{h_0}{2}} + e^{-\frac{h_0}{2}})^2}.$$

Hence

$$c_{ij} = c_{[N]}(j) = \sup_{x=\hat{x} \text{ off } j} \left| \prod_{i=1}^{N} \Psi(x^i \mid x) - \prod_{i=1}^{N} \Psi(\hat{x}^i \mid \hat{x}) \right| \leq \sum_{i \neq j} \frac{2\beta |W_{ij}|}{(e^{\frac{h_0}{2}} + e^{-\frac{h_0}{2}})^2} + \frac{2\beta \sum_{k \neq j} |W_{jk}| + 2|h|}{(e^{\frac{h_0}{2}} + e^{-\frac{h_0}{2}})^2}$$

$$= \frac{2\beta(\sum_{i \neq j} |W_{ij}| + \sum_{k \neq j} |W_{jk}|) + 2|h|}{(e^{\frac{|h|}{4}} + e^{-\frac{|h|}{4}})^2} \leq \frac{4|h|}{(e^{\frac{|h|}{4}} + e^{-\frac{|h|}{4}})^2}.$$

Finally,

$$\|C\|_\infty = \sup_i \sum_{j \in [N]} c_i(j) \leq \frac{4|h|N}{(e^{\frac{|h|}{4}} + e^{-\frac{|h|}{4}})^2}.$$

Combine with Lemma B.2, we have

$$\text{Var}_{\nu_{\beta,h}}(f) \leq \frac{1}{1 - \frac{4N|h|}{(e^{\frac{|h|}{4}} + e^{-\frac{|h|}{4}})^2}} \mathcal{E}(f,f).$$

## C.8. Chernoff-type Error Bounds for DMCMC Samplers

This section details the derivation of Chernoff-type error bounds for the Monte Carlo estimators based on DMCMC samplers, expanding upon the theoretical results presented in Section 4. These bounds quantify the concentration of the sample average around the true posterior mean. For a sequence of samples $\{X_1, X_2, \ldots, X_n\}$ generated by a DMCMC chain, we establish the following corollary:

**Corollary C.2.** *Let $X_1 \sim q$ denote the initial distribution of the samples. Under the assumption that the Gibbs measure (13) satisfies the large field Condition 4.1, for any $\varepsilon > 0$, the following Chernoff-type error bound holds:*

$$P_q\left[\left|\frac{1}{n}\sum_{i=1}^{n} X_i - \mathbb{E}_{\nu_{\beta,h}}[X]\right| \geq \varepsilon\right] \leq C_{\gamma_{gap},n,q}e^{-\frac{n\varepsilon^2\gamma_{gap}}{c}},$$

*where $c$ is an absolute constant, $C_{\gamma_{gap},n,q}$ is a rational function, and the spectral gap $\gamma_{gap}$ is specific to each DMCMC algorithm:*

- *For classical Metropolis chains, $\gamma_{gap} = 1 - 2|h|e^{-|h|}$;*

- *For the single-site gradient-informed MH algorithm, $\gamma_{gap} = 1 - \frac{|h|}{(e^{\frac{|h|}{4}}+e^{-\frac{|h|}{4}})^2}$;*

- *For DULA, $\gamma_{gap} = 1 - \frac{4|h|N}{(e^{\frac{|h|}{4}}+e^{-\frac{|h|}{4}})^2}$.*

*Proof.* Our approach leverages general results on the concentration of measure for Markov chains. A Metropolis-Hastings (MH) kernel $P(x'|x)$ is typically defined by:

$$P(x'|x) = \Psi(x'|x)\min\left\{1, \frac{\nu_{\beta,h}(x')\Psi(x|x')}{\nu_{\beta,h}(x)\Psi(x'|x)}\right\},$$

where $\Psi(\cdot|\cdot)$ represents the proposal distribution. A key property of these MH kernels is their reversibility with respect to the target Gibbs measure $\nu_{\beta,h}$. Furthermore, given that we are sampling from a finite state space $\{-1,1\}^N$, these MH algorithms are inherently irreducible.

These properties of reversibility and irreducibility are precisely the conditions required for Lemma C.1 to apply. Lemma C.1 states that for a reversible and irreducible Markov chain, a Chernoff-type error bound of the form

$$P_q\left[\left|\frac{1}{n}\sum_{i=1}^{n} X_i - \mathbb{E}_{\nu_{\beta,h}}[X]\right| \geq \varepsilon\right] \leq C_{\gamma_{gap},n,q}e^{-\frac{n\varepsilon^2\gamma_{gap}}{c}}$$

holds, where $\gamma_{gap}$ is the spectral gap of the underlying Markov chain.

Finally, by incorporating the specific spectral gap calculations derived for each DMCMC variant—classical Metropolis chains (Theorem 4.5), single-site gradient-informed MH (Theorem 4.8), and DULA (Theorem 4.9)—we substitute their respective $\gamma_{gap}$ values into this general bound, thus completing the proof. $\square$

# D. Experimental Miscellaneous

In this section, we first benchamrk the QUBO instances in section D.1 and provide detailed experimental settings for reproducibility in Section D.2. We then present additional experimental results in Section D.3 to further validate our framework's effectiveness. Finally, in Section D.4, we conduct comprehensive ablation studies on various components of our framework, such as the $\alpha$-schedule in SL and the influence of hyperparameters on sampling performance and convergence properties.

## D.1. QUBO Instances

In this section, we describe the form of Binary Quadratic Distributions for the QUBO instances used in our experiments. For instances with $x \in \{0,1\}^N$, we transform them to $\{-1,1\}^N$ through the mapping $y = \frac{x+1}{2}$ in our experimental setup.

**Maximum Independent Set** Given a graph $G = (V, E)$ with $N$ nodes, the Maximum Independent Set (MIS) problem can be formulated as:

$$\max_{x \in 0,1^N} \sum_{i=1}^{N} -c_i x_i \text{ subject to } x_i x_j = 0 \text{ for all } (i, j) \in E$$

which can be transformed into a BQD:

$$p(x) \propto e^{-\frac{\beta}{2}\left(-c^T x + \lambda \frac{x^T A x}{2}\right)}$$

where $A$ is the adjacency matrix, $\beta$ is the inverse temperature and $\lambda$ is the penalty coefficient ensuring constraint satisfaction. Following DISCS (Goshvadi et al., 2024), we set $c = 1$, $\lambda = 1.0001$ and use their annealing schedule and post-processing method to obtain feasible solutions. The evaluation is conducted on five datasets: four from Erdős-Rényi random graphs ER-[700-800] with densities {0.05, 0.10, 0.15, 0.25} and one from SATLIB.

**Maximum Cut** The Maximum Cut (MaxCut) problem on a graph $G = (V, E)$ with $D$ nodes can be formulated as:

$$\max_{x \in \{-1,1\}^N} \sum_{(i,j) \in E} w_{ij}\left(\frac{1 - x_i x_j}{2}\right)$$

which corresponds to the following BQD:

$$p(x) \propto e^{-\beta\left(\sum_{(i,j) \in E} w_{ij}\left(\frac{1 - x_i x_j}{2}\right)\right)}$$

where $w_{ij} = 1$ in the experiments. Following DISCS (Goshvadi et al., 2024), we evaluate on seven datasets: three from different sizes of Erdős-Rényi random graphs with the density 0.15 (ER-256-300, ER-512-600, ER-1024-1100), three from different sizes of Barabási-Albert random graphs (ba-256-300, ba-512-600, ba-1024-1100), and one from Optsicom.

**Maximum Clique** The Maximum Clique problem on a graph $G = (V, E)$ with $D$ nodes can be formulated as:

$$\min_{x \in \{0,1\}^N} -\sum_{i=1}^{N} c_i x_i \text{ subject to } x_i x_j = 0 \text{ for all } (i, j) \notin E$$

which corresponds to the following BQD:

$$p(x) \propto e^{-\beta\left(-c^T x + \frac{\lambda}{2}(\mathbf{1}^T x \cdot (\mathbf{1}^T x - \mathbf{1}) - x^T A x)\right)}$$

Following DISCS (Goshvadi et al., 2024), we set $c = 1$, $\lambda = 1.0001$ and maintain the same post-processing method. We evaluate on two datasets: Twitter and RB.

### D.2. Experimental Setting

In this part, we provide detailed experimental settings for the configuration of three key components in our framework: the discrete MCMC sampler settings, the Stochastic Localization algorithm parameters, and the exponential decay step allocation strategy.

**Discrete MCMC Sampler Settings** We employ three different discrete MCMC samplers in our experiments: Gibbs with Gradients (GWG) (Grathwohl et al., 2021), Path Auxiliary Sampler (PAS) (Sun et al., 2021), and Discrete Metropolis-Adjusted Langevin Algorithm (DMALA) (Zhang et al., 2022), for all samplers, we use adaptive step sizes with a target acceptance rate of 0.574 and balancing function of $g(t) = \sqrt{t}$. The specific configurations are as follows: For GWG and PAS, we configure it to flip one bit at each step. For DMALA, we set the initial step size to 0.2, schedule the step size adjustment every 100 steps, and reset the gradient estimate every 20 steps.

**Exponential Decay Step Allocation** Given the total number of SL SDE iterations $K$ and total MCMC steps $N_{\text{tot}}$ (10,000 in our work), we need to allocate $N_t$ MCMC steps for posterior estimation at each time step $t$, ensuring $\sum_{t=1}^{K} N_t = N_{\text{tot}}$. Our exponential decay allocation strategy works as follows: First, we assign a minimum number of steps $N_{\min}$ to each time step. Then, we distribute the remaining steps $N_{\text{tot}} - K N_{\min}$ exponentially according to $N_t = N_{\min} + c \cdot r^{t/K}$, where $r$ is the decay rate and $c$ is a normalization constant ensuring the sum constraint. To handle discretization effects, we floor the continuous schedule to integers and carefully distribute the remaining steps to maintain the exact total.

*Table 4.* Results of ablative experiments for MIS ER-0.05 and MIS ER-0.10.

| Ablation | SL-GWG | SL-PAS | SL-DMALA | Ablation | SL-GWG | SL-PAS | SL-DMALA |
|---|---|---|---|---|---|---|---|
| CLASSIC | $103.69 \pm 1.57$ | $104.31 \pm 1.79$ | $103.06 \pm 1.82$ | CLASSIC | $60.25 \pm 0.79$ | $61.63 \pm 0.96$ | $60.94 \pm 1.06$ |
| GEOM(1,1) | $103.97 \pm 1.55$ | $104.19 \pm 1.67$ | $103.41 \pm 1.85$ | GEOM(1,1) | $61.47 \pm 1.06$ | $61.75 \pm 0.83$ | $60.88 \pm 0.96$ |
| GEOM(2,1) | $104.37 \pm 1.52$ | $104.53 \pm 1.48$ | $103.59 \pm 1.80$ | GEOM(2,1) | $61.94 \pm 0.93$ | $61.91 \pm 0.88$ | $61.34 \pm 0.96$ |
| K=256 | $104.19 \pm 1.65$ | $104.53 \pm 1.48$ | $103.34 \pm 1.57$ | K=256 | $61.41 \pm 0.82$ | $61.91 \pm 0.88$ | $60.78 \pm 0.99$ |
| K=512 | $104.19 \pm 1.65$ | $104.19 \pm 1.72$ | $103.59 \pm 1.80$ | K=512 | $61.94 \pm 0.93$ | $61.78 \pm 0.96$ | $60.94 \pm 1.00$ |
| K=1024 | $104.37 \pm 1.52$ | $104.22 \pm 1.49$ | $103.03 \pm 1.69$ | K=1024 | $61.53 \pm 0.83$ | $61.69 \pm 0.95$ | $61.34 \pm 0.96$ |
| $\sigma$=1 | $103.25 \pm 1.70$ | $104.16 \pm 1.42$ | $103.19 \pm 1.74$ | $\sigma$=1 | $61.47 \pm 0.83$ | $61.69 \pm 0.77$ | $61.09 \pm 1.16$ |
| $\sigma$=5 | $103.81 \pm 1.63$ | $104.22 \pm 1.67$ | $103.34 \pm 1.61$ | $\sigma$=5 | $61.69 \pm 1.04$ | $61.69 \pm 0.88$ | $60.81 \pm 1.04$ |
| $\sigma$=10 | $104.00 \pm 1.54$ | $104.34 \pm 1.81$ | $103.28 \pm 1.64$ | $\sigma$=10 | $61.66 \pm 0.92$ | $61.50 \pm 0.83$ | $61.22 \pm 1.05$ |

**SL Settings** For the alpha-schedule in Stochastic Localization, we adopt the GEOM(2,1) schedule $\alpha(t) = t(1-t)^{-1/2}$ as suggested in prior works. We use uniform time discretization and employ exponential integrator for SDE integration. When performing posterior estimation in SL, we use the same MCMC sampler configurations as in their standalone sampling counterparts. Other hyperparameters are determined through extensive hyperparameter optimization. Specifically, we tune the following parameters: the number of discretization steps $K \in \{256, 512, 1024\}$, initial and final noise scales $\epsilon, \epsilon_{\mathrm{end}} \in [10^{-4}, 10^{-1}]$, MCMC sample ratio (proportion of last samples used for posterior estimation) in $[0.1, 1]$ with step 0.1, MCMC step exponential decay rate $r$ in $[10^{-4}, 10^{-1}]$, minimum MCMC steps $N_{\mathrm{min}} \in \{2, 4, 6, 8, 16, 32\}$, and noise parameter $\sigma \in \{1, 2, 4, 6, 8, 10, 15, 20\}$. The total number of MCMC steps is fixed at 10,000. The optimal hyperparameters found through our search will be released with our open-source code.

All experiments are conducted on a single NVIDIA A40 GPU and an AMD EPYC 7513 CPU. The code is available at https://github.com/LOGO-CUHKSZ/SLDMCMC.

### D.3. Further Experimental Results

To provide a more comprehensive understanding of our framework's behavior, we visualize the sampling trajectories for both the base MCMC samplers (DMALA, GWG, and PAS) and the SL-enhanced framework across different test cases. The shaded areas in the trajectories, representing the variance across multiple instances, are generally narrow, indicating the stability of all methods. Most notably, SL-based samplers (solid lines) and their base MCMC counterparts (dashed lines) achieve comparable final performance across all test cases, demonstrating the reliability of both approaches.

### D.4. Further Ablations

Based on the comprehensive ablation experiments shown in Tables 4 - 9, we analyze the impact of three key components in SL: the $\alpha$-schedule, number of SL iterations K, and noise parameter $\sigma$.

First, regarding the $\alpha$-schedule, we adopt the geometric schedule GEOM($\alpha_1$, $\alpha_2$) from SLIPS (Grenioux et al., 2024), defined as GEOM($\alpha_1$, $\alpha_2$)$(t) = t^{\frac{\alpha_1}{2}}(1-t)^{-\frac{\alpha_2}{2}}$, where $t \in [0, 1]$ and $\alpha_1 \geq 1$, $\alpha_2 > 0$. Both GEOM(1,1) and GEOM(2,1) demonstrate superior performance over the CLASSIC schedule (defined as $\alpha(t) = t$) in MIS problems. However, this impact becomes negligible in MaxCut instances, where all schedules perform similarly with variations within 0.1%.

Second, the number of SL iterations K shows a problem-dependent sensitivity pattern. While K exhibits significant influence on performance in MIS tasks, its impact becomes minimal in MaxCut problems where the performance differences among K=256, 512, and 1024 are statistically insignificant ($\leq$0.01% variation).

Third, the noise parameter $\sigma$ demonstrates similar problem-specific characteristics. In MIS instances, moderate noise levels ($\sigma = 5$) generally yield optimal performance with notable improvements over other settings. However, in MaxCut problems, the choice of $\sigma$ has minimal impact on the final performance, with variations typically less than 0.05%.

When comparing methods across problem types, SL-PAS exhibits superior performance in MIS tasks compared to SL-GWG and SL-DMALA. However, this advantage diminishes in MaxCut instances, where all three methods achieve nearly identical results.

*Table 5.* Results of ablative experiments for MIS ER-0.20 and MIS ER-0.25.

| Ablation | SL-GWG | SL-PAS | SL-DMALA | Ablation | SL-GWG | SL-PAS | SL-DMALA |
|---|---|---|---|---|---|---|---|
| CLASSIC | $34.06 \pm 0.61$ | $33.91 \pm 0.58$ | $33.03 \pm 0.77$ | CLASSIC | $27.88 \pm 0.54$ | $26.50 \pm 0.56$ | $27.31 \pm 0.58$ |
| GEOM(1,1) | $34.09 \pm 0.63$ | $34.06 \pm 0.50$ | $33.97 \pm 0.64$ | GEOM(1,1) | $27.56 \pm 0.50$ | $27.84 \pm 0.51$ | $27.44 \pm 0.50$ |
| GEOM(2,1) | $34.38 \pm 0.74$ | $34.38 \pm 0.54$ | $34.00 \pm 0.56$ | GEOM(2,1) | $28.00 \pm 0.56$ | $28.03 \pm 0.59$ | $27.75 \pm 0.66$ |
| K=256 | $34.19 \pm 0.73$ | $34.25 \pm 0.61$ | $33.84 \pm 0.62$ | K=256 | $27.84 \pm 0.57$ | $27.94 \pm 0.50$ | $27.41 \pm 0.55$ |
| K=512 | $34.03 \pm 0.53$ | $34.09 \pm 0.52$ | $34.00 \pm 0.56$ | K=512 | $27.88 \pm 0.48$ | $27.91 \pm 0.63$ | $27.47 \pm 0.56$ |
| K=1024 | $34.38 \pm 0.74$ | $34.38 \pm 0.54$ | $33.72 \pm 0.84$ | K=1024 | $28.00 \pm 0.56$ | $28.03 \pm 0.59$ | $27.75 \pm 0.66$ |
| $\sigma$=1 | $33.72 \pm 0.57$ | $33.69 \pm 0.58$ | $33.84 \pm 0.67$ | $\sigma$=1 | $27.81 \pm 0.53$ | $27.72 \pm 0.51$ | $27.06 \pm 0.50$ |
| $\sigma$=5 | $34.22 \pm 0.74$ | $34.00 \pm 0.66$ | $33.81 \pm 0.68$ | $\sigma$=5 | $27.72 \pm 0.57$ | $27.94 \pm 0.43$ | $27.47 \pm 0.50$ |
| $\sigma$=10 | $34.00 \pm 0.43$ | $34.22 \pm 0.54$ | $33.88 \pm 0.65$ | $\sigma$=10 | $27.88 \pm 0.65$ | $27.94 \pm 0.61$ | $27.75 \pm 0.66$ |

*Table 6.* Results of ablative experiments for MIS SATLib and MaxClique RB.

| Ablation | SL-GWG | SL-PAS | SL-DMALA | Ablation | SL-GWG | SL-PAS | SL-DMALA |
|---|---|---|---|---|---|---|---|
| CLASSIC | $418.93 \pm 14.39$ | $419.35 \pm 14.37$ | $415.23 \pm 14.51$ | CLASSIC | $87.52 \pm 6.18$ | $87.59 \pm 6.19$ | $87.16 \pm 6.11$ |
| GEOM(1,1) | $419.03 \pm 14.31$ | $419.61 \pm 14.37$ | $415.72 \pm 14.48$ | GEOM(1,1) | $87.52 \pm 6.14$ | $87.43 \pm 6.11$ | $87.10 \pm 6.09$ |
| GEOM(2,1) | $419.16 \pm 14.39$ | $419.71 \pm 14.35$ | $415.72 \pm 14.37$ | GEOM(2,1) | $87.60 \pm 6.16$ | $87.65 \pm 6.14$ | $87.31 \pm 6.15$ |
| K=256 | $419.03 \pm 14.32$ | $419.71 \pm 14.35$ | $415.57 \pm 14.55$ | K=256 | $87.43 \pm 6.16$ | $87.65 \pm 6.14$ | $87.19 \pm 6.08$ |
| K=512 | $419.16 \pm 14.39$ | $419.62 \pm 14.37$ | $415.72 \pm 14.37$ | K=512 | $87.42 \pm 6.09$ | $87.41 \pm 6.13$ | $87.18 \pm 6.06$ |
| K=1024 | $419.03 \pm 14.45$ | $419.59 \pm 14.35$ | $415.66 \pm 14.52$ | K=1024 | $87.60 \pm 6.16$ | $87.56 \pm 6.17$ | $87.31 \pm 6.15$ |
| $\sigma$=1 | $418.85 \pm 14.31$ | $419.51 \pm 14.33$ | $415.43 \pm 14.29$ | $\sigma$=1 | $86.77 \pm 6.11$ | $87.35 \pm 6.13$ | $87.08 \pm 6.11$ |
| $\sigma$=5 | $419.00 \pm 14.37$ | $419.55 \pm 14.35$ | $415.70 \pm 14.39$ | $\sigma$=5 | $87.42 \pm 6.18$ | $87.55 \pm 6.20$ | $87.26 \pm 6.16$ |
| $\sigma$=10 | $419.01 \pm 14.39$ | $419.71 \pm 14.35$ | $415.72 \pm 14.37$ | $\sigma$=10 | $87.42 \pm 6.21$ | $87.45 \pm 6.17$ | $87.31 \pm 6.15$ |

# E. Computational Complexity Analysis

We analyze the computational complexity of the Stochastic Localization (SL) framework combined with Discrete MCMC (DMCMC) samplers, compared to using DMCMC alone. We consider the total operational cost in terms of fundamental operations like function evaluations, pseudo-gradient calculations, or matrix operations, as these often dominate the runtime for the target distributions we consider. Let $N$ be the dimension of the problem (number of binary variables), $T$ be the total number of SL iterations, and $M$ be the total budget of DMCMC steps across all SL iterations.

### E.1. Baseline DMCMC Complexity

For standard DMCMC methods used as baselines in our study, such as Gibbs With Gradients (GWG) (Grathwohl et al., 2021), Path Auxiliary Sampler (PAS) (Sun et al., 2021), and Discrete Metropolis-adjusted Langevin Algorithm (DMALA) (Zhang et al., 2022), the dominant computational cost per MCMC step typically involves evaluating the target distribution's pseudo-gradient or related quantities. For many relevant models, including the Binary Quadratic Distribution (BQD) studied in this paper, these operations scale at least quadratically with the dimension, $O(N^2)$. Given a total budget of $M$ MCMC steps, the overall complexity of a baseline DMCMC method is thus $O(MN^2)$.

### E.2. SL + DMCMC Complexity

When using SL with DMCMC, the total MCMC budget $M$ is distributed across $T$ SL iterations. The core cost associated with performing $M$ DMCMC steps remains $O(MN^2)$ in total. In addition to the MCMC sampling, each of the $T$ SL iterations involves a few key operations: **(1)** MC estimation of the conditional mean: This step involves drawing samples from the current posterior distribution and computing their mean. If we draw a fixed, relatively small number of samples per SL iteration, this calculation takes $O(N)$ time per iteration as it involves vector operations in $\mathbb{R}^N$. **(2)** SDE simulation: Updating the localization variable $Y_t^\alpha$ according to the SDE involves vector additions and scaling, which takes $O(N)$ time per iteration. The total computational overhead introduced by the SL process itself, over $T$ iterations, is therefore $O(T \times (N + N)) = O(TN)$. Combining the cost of the total MCMC steps and the SL overhead, the total computational complexity of the SL+DMCMC framework is $O(MN^2 + TN)$.

*Table 7.* Results of ablative experiments for MaxCut ER 256-300 and MaxCut BA 256-300.

| Ablation | SL-GWG | SL-PAS | SL-DMALA | Ablation | SL-GWG | SL-PAS | SL-DMALA |
|---|---|---|---|---|---|---|---|
| CLASSIC | $101.92 \pm 1.76$ | $101.91 \pm 1.76$ | $101.92 \pm 1.76$ | CLASSIC | $99.96 \pm 0.07$ | $99.97 \pm 0.06$ | $99.95 \pm 0.07$ |
| GEOM(1,1) | $101.92 \pm 1.76$ | $101.92 \pm 1.76$ | $101.92 \pm 1.76$ | GEOM(1,1) | $99.97 \pm 0.06$ | $99.97 \pm 0.06$ | $99.95 \pm 0.08$ |
| GEOM(2,1) | $101.92 \pm 1.76$ | $101.92 \pm 1.76$ | $101.92 \pm 1.76$ | GEOM(2,1) | $99.97 \pm 0.05$ | $99.98 \pm 0.05$ | $99.96 \pm 0.06$ |
| K=256 | $101.91 \pm 1.76$ | $101.92 \pm 1.76$ | $101.91 \pm 1.76$ | K=256 | $99.96 \pm 0.07$ | $99.97 \pm 0.06$ | $99.96 \pm 0.07$ |
| K=512 | $101.92 \pm 1.76$ | $101.92 \pm 1.76$ | $101.92 \pm 1.76$ | K=512 | $99.97 \pm 0.06$ | $99.97 \pm 0.06$ | $99.96 \pm 0.07$ |
| K=1024 | $101.92 \pm 1.76$ | $101.92 \pm 1.76$ | $101.92 \pm 1.76$ | K=1024 | $99.97 \pm 0.06$ | $99.97 \pm 0.05$ | $99.96 \pm 0.06$ |
| $\sigma$=1 | $101.90 \pm 1.76$ | $101.89 \pm 1.76$ | $101.90 \pm 1.76$ | $\sigma$=1 | $99.94 \pm 0.08$ | $99.95 \pm 0.07$ | $99.81 \pm 0.12$ |
| $\sigma$=5 | $101.92 \pm 1.76$ | $101.92 \pm 1.76$ | $101.92 \pm 1.76$ | $\sigma$=5 | $99.97 \pm 0.06$ | $99.97 \pm 0.05$ | $99.95 \pm 0.07$ |
| $\sigma$=10 | $101.92 \pm 1.76$ | $101.92 \pm 1.76$ | $101.92 \pm 1.76$ | $\sigma$=10 | $99.97 \pm 0.06$ | $99.97 \pm 0.06$ | $99.96 \pm 0.07$ |

*Table 8.* Results of ablative experiments for MaxCut ER 512-600 and MaxCut ER 1024-1100.

| Ablation | SL-GWG | SL-PAS | SL-DMALA | Ablation | SL-GWG | SL-PAS | SL-DMALA |
|---|---|---|---|---|---|---|---|
| CLASSIC | $100.16 \pm 0.12$ | $100.17 \pm 0.12$ | $100.14 \pm 0.13$ | CLASSIC | $100.08 \pm 0.14$ | $100.12 \pm 0.14$ | $100.06 \pm 0.14$ |
| GEOM(1,1) | $100.16 \pm 0.12$ | $100.17 \pm 0.12$ | $100.14 \pm 0.13$ | GEOM(1,1) | $100.10 \pm 0.14$ | $100.12 \pm 0.13$ | $100.07 \pm 0.14$ |
| GEOM(2,1) | $100.16 \pm 0.12$ | $100.17 \pm 0.12$ | $100.14 \pm 0.12$ | GEOM(2,1) | $100.10 \pm 0.13$ | $100.12 \pm 0.14$ | $100.08 \pm 0.14$ |
| K=256 | $100.15 \pm 0.12$ | $100.16 \pm 0.12$ | $100.14 \pm 0.12$ | K=256 | $100.09 \pm 0.14$ | $100.12 \pm 0.14$ | $100.08 \pm 0.14$ |
| K=512 | $100.16 \pm 0.12$ | $100.17 \pm 0.12$ | $100.14 \pm 0.12$ | K=512 | $100.09 \pm 0.14$ | $100.12 \pm 0.14$ | $100.07 \pm 0.14$ |
| K=1024 | $100.16 \pm 0.12$ | $100.17 \pm 0.12$ | $100.14 \pm 0.12$ | K=1024 | $100.10 \pm 0.14$ | $100.12 \pm 0.14$ | $100.07 \pm 0.14$ |
| $\sigma$=1 | $100.15 \pm 0.12$ | $100.16 \pm 0.12$ | $100.11 \pm 0.13$ | $\sigma$=1 | $100.07 \pm 0.14$ | $100.11 \pm 0.13$ | $100.07 \pm 0.14$ |
| $\sigma$=5 | $100.16 \pm 0.12$ | $100.17 \pm 0.12$ | $100.14 \pm 0.13$ | $\sigma$=5 | $100.10 \pm 0.14$ | $100.12 \pm 0.13$ | $100.07 \pm 0.14$ |
| $\sigma$=10 | $100.16 \pm 0.12$ | $100.17 \pm 0.12$ | $100.14 \pm 0.13$ | $\sigma$=10 | $100.09 \pm 0.14$ | $100.12 \pm 0.14$ | $100.08 \pm 0.14$ |

## E.3. Comparison

Comparing the total complexity $O(MN^2 + TN)$ with the baseline $O(MN^2)$, the SL framework adds an $O(TN)$ overhead. In the high-dimensional settings where advanced MCMC methods are typically applied ($N$ is large), and with practical choices for the number of SL iterations $T$ relative to the total MCMC budget $M$, this overhead is often negligible. For example, in our experiments, we used $T \in \{256, 512, 1024\}$ while the total MCMC steps $M$ went up to $10,000$. In such scenarios, $TN$ is much smaller than $MN^2$. Thus, the SL framework allows for potential improvements in mixing and convergence properties by transforming the target distribution towards an easier-to-sample form over time, without introducing a prohibitive increase in computational cost compared to the baseline DMCMC methods.

To validate our theoretical complexity analysis, we measured the actual running times across various benchmark problems, as shown in Tables 10 and 11. The results largely confirm our theoretical expectations. For MIS problems, the SL variants show a small overhead (typically 1-6%) compared to their base counterparts. This slight increase is consistent with our complexity analysis predicting an additional $O(TN)$ term. In some instances (e.g., MaxClique RB and TWITTER, and some larger MaxCut instances), the SL framework appears marginally faster than the base methods. However, these apparent improvements should be attributed to hardware-specific factors, implementation details, and measurement variability. Such minor variations are common in empirical runtime measurements and do not contradict our theoretical analysis. Overall, the empirical results demonstrate that the SL framework introduces only marginal computational overhead in practice, which aligns with our theoretical analysis where $TN \ll MN^2$ in high-dimensional settings.

Table 9. Results of ablative experiments for MaxCut BA 512-600 and MaxCut BA 1024-1100.

| Ablation | SL-GWG | SL-PAS | SL-DMALA | Ablation | SL-GWG | SL-PAS | SL-DMALA |
|---|---|---|---|---|---|---|---|
| CLASSIC | $100.87 \pm 0.48$ | $100.92 \pm 0.48$ | $100.77 \pm 0.49$ | CLASSIC | $101.53 \pm 0.38$ | $101.66 \pm 0.38$ | $101.21 \pm 0.39$ |
| GEOM(1,1) | $100.88 \pm 0.48$ | $100.92 \pm 0.48$ | $100.78 \pm 0.48$ | GEOM(1,1) | $101.56 \pm 0.37$ | $101.66 \pm 0.37$ | $101.23 \pm 0.39$ |
| GEOM(2,1) | $100.89 \pm 0.49$ | $100.92 \pm 0.48$ | $100.79 \pm 0.49$ | GEOM(2,1) | $101.55 \pm 0.37$ | $101.66 \pm 0.37$ | $101.23 \pm 0.38$ |
| K=256 | $100.87 \pm 0.49$ | $100.92 \pm 0.48$ | $100.78 \pm 0.49$ | K=256 | $101.55 \pm 0.38$ | $101.66 \pm 0.37$ | $101.22 \pm 0.38$ |
| K=512 | $100.88 \pm 0.49$ | $100.92 \pm 0.48$ | $100.79 \pm 0.49$ | K=512 | $101.56 \pm 0.38$ | $101.66 \pm 0.37$ | $101.23 \pm 0.38$ |
| K=1024 | $100.89 \pm 0.49$ | $100.92 \pm 0.48$ | $100.78 \pm 0.49$ | K=1024 | $101.56 \pm 0.38$ | $101.67 \pm 0.37$ | $101.21 \pm 0.39$ |
| $\sigma$=1 | $100.77 \pm 0.49$ | $100.90 \pm 0.48$ | $100.76 \pm 0.49$ | $\sigma$=1 | $101.31 \pm 0.37$ | $101.63 \pm 0.37$ | $101.16 \pm 0.39$ |
| $\sigma$=5 | $100.88 \pm 0.48$ | $100.92 \pm 0.49$ | $100.78 \pm 0.49$ | $\sigma$=5 | $101.55 \pm 0.37$ | $101.66 \pm 0.37$ | $101.22 \pm 0.39$ |
| $\sigma$=10 | $100.88 \pm 0.49$ | $100.92 \pm 0.48$ | $100.78 \pm 0.49$ | $\sigma$=10 | $101.55 \pm 0.37$ | $101.67 \pm 0.37$ | $101.22 \pm 0.39$ |

Table 10. Running time (in seconds) on MIS and MaxClique benchmarks for various methods and their SL variants. Bold values indicate faster performance between paired methods.

| Method | MIS | | | | | MaxClique | |
|---|---|---|---|---|---|---|---|
| | ER-0.05 | ER-0.10 | ER-0.20 | ER-0.25 | SATLIB | RB | TWITTER |
| DMALA | **23.873** | **35.511** | **77.866** | **103.636** | **188.901** | 3025.712 | 120.634 |
| SL-DMALA | 24.997 | 37.621 | 78.494 | 104.367 | 200.743 | **3013.018** | **117.597** |
| PAS | **32.648** | **44.558** | **86.655** | **112.305** | **377.038** | 3049.668 | 125.222 |
| SL-PAS | 36.134 | 49.425 | 90.428 | 115.630 | 388.979 | **3041.619** | **125.502** |
| GWG | **27.249** | **37.743** | **79.305** | **104.461** | **326.739** | **3030.762** | **117.964** |
| SL-GWG | 28.056 | 40.556 | 81.412 | 106.017 | 330.687 | 3031.967 | 119.806 |

Table 11. Running time (in seconds) on MaxCut benchmarks for various methods and their SL variants. Bold values indicate faster performance between paired methods.

| Method | ER | | | BA | | | OPTSICOM |
|---|---|---|---|---|---|---|---|
| | 256-300 | 512-600 | 1024-1100 | 256-300 | 512-600 | 1024-1100 | |
| DMALA | **167.341** | **755.462** | 2814.459 | **75.166** | **141.391** | **271.073** | **4.772** |
| SL-DMALA | 169.672 | 759.201 | **2807.121** | 81.436 | 148.428 | 284.511 | 7.459 |
| PAS | **196.667** | **815.633** | 2873.359 | **133.742** | **267.957** | **591.858** | **6.647** |
| SL-PAS | 201.665 | 820.853 | **2869.082** | 140.680 | 279.453 | 619.575 | 10.487 |
| GWG | **178.039** | **780.655** | 2943.244 | 714.469 | 2749.896 | 7872.727 | **6.476** |
| SL-GWG | 181.171 | 784.497 | **2936.596** | **710.698** | **2695.718** | **6802.453** | 9.146 |

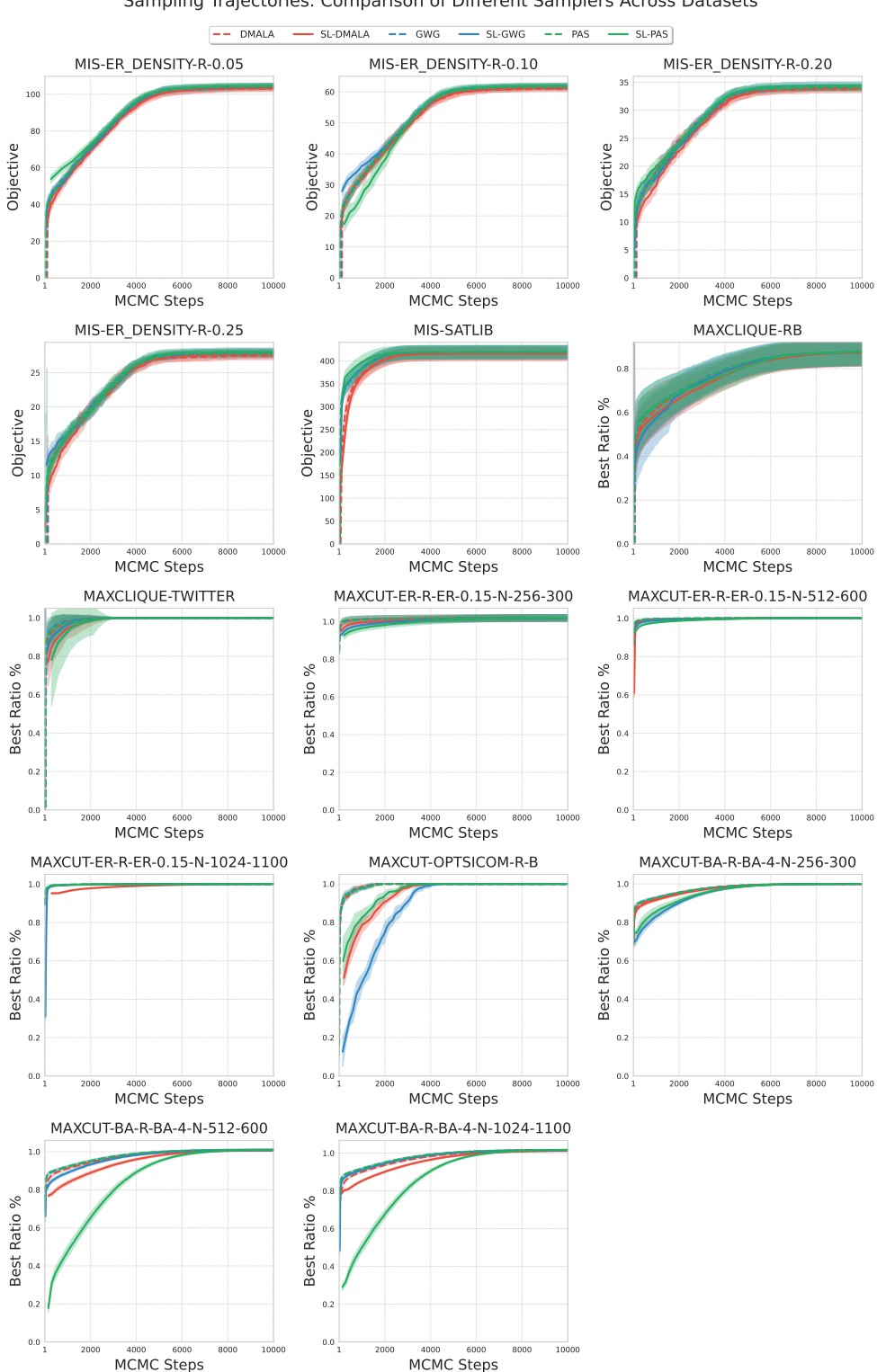

*Figure 2.* Sampling trajectories comparison between MCMC samplers (dash lines) and their SL-based counterparts (solid lines) across different datasets. The shaded areas in the trajectories, representing the variance across multiple instances

