# OpenReview forum: "Sampling from Binary Quadratic Distributions via Stochastic Localization"
_ICML.cc/2025/Conference — ICML 2025 poster_

### Official Review · Reviewer_mA2w · 2025-03-13

**Overall Recommendation:** 3

**Summary:**

This work addresses the problem of sampling from binary quadratic distributions. The authors apply a stochastic localization framework and focus on a key component—the counting/expectation of the posterior distribution. To this end, they establish Poincaré inequalities for the posterior, from which they derive a spectral gap. Experiments further demonstrate that stochastic localization consistently improves sampling efficiency.

**Claims And Evidence:**

The introduction seems to overstate the novelty of the work by emphasizing the use of stochastic localization in binary quadratic distributions, which has already been widely applied in discrete sampling. The primary theoretical contribution of the paper is the quantification of Poincaré inequalities for the posterior distributions.

**Essential References Not Discussed:**

No, the paper appears to cite all the essential prior works. It covers key contributions in discrete MCMC sampling and stochastic localization, which are sufficient for understanding its context and contributions.

**Experimental Designs Or Analyses:**

I am less familiar with the experimental aspects, so I only conducted a high-level review. Based on my assessment, the experimental design appears reasonable and generally supportive of the claims.

**Methods And Evaluation Criteria:**

Yes, the proposed methods and evaluation criteria are well-chosen. The framework leverages rigorous theoretical analysis (e.g., Poincaré inequalities and spectral gap bounds) tailored for discrete MCMC samplers, while the benchmark datasets (from common combinatorial optimization problems) effectively capture the challenges of sampling from binary quadratic distributions.

**Other Comments Or Suggestions:**

It would better to question in Line 66 as "Can SL reduce sampling difficulty in the binary quadratic distributions, as it does in continuous settings, by constructing easily samplable posterior distributions?" since there is various SL methods for discrete sampling tasks as discussed in Appendix A.

**Other Strengths And Weaknesses:**

The theoretical guarantees rely on a strong external field assumption that may not hold in all practical scenarios, potentially limiting the generality of the results.

**Questions For Authors:**

- Could you provide the convergence rate/query complexity/iteration complexity explicitly? and compare it with existing results?

**Relation To Broader Scientific Literature:**

I don't know.

**Theoretical Claims:**

I don't find problems.

---

> ### Author Rebuttal · Authors · 2025-03-31
>
> We sincerely thank the reviewer for these insightful comments.
> > Overstating the novelty by emphasizing the use of SL in binary quadratic distributions
>
> We appreciate the feedback on framing. While SL *concepts* have appeared in discrete settings (as discussed in Appendix A), prior works often focus on specific models (e.g., SK, Ising with specific structures) and employ model-specific techniques or assumptions. Our contribution lies in:
> - Developing an SL framework using **standard, general-purpose DMCMC samplers** for the posterior estimation step.
> - Providing **theoretical guarantees for general BQDs** without requiring additional model-specific structure beyond the quadratic form. This makes our analysis broadly applicable.
>
> We agree our main theoretical novelty is the rigorous quantification of Poincaré inequalities for the posterior distributions within this general BQD setting. We will **revise the introduction** to state this more clearly and accurately reflect the relationship to prior discrete SL work, emphasizing the generality of our approach and theory.
>
> > Satisfiability of assumption 4.1
>
> As detailed in Remark 4.2, the external field $h$ in the posterior (Eq. 12) is $b+\frac{\alpha(t)Y_t}{\sigma^2t}$. Theorem 3.1 and the SL construction ensure that $|h|$ grows large as $t$ increases **with high probability** (explained around line 161). Therefore, Assumption 4.1 is **not a restrictive assumption on the problem instance, but rather a condition that holds naturally as a consequence of the SL dynamics** for sufficiently large $t$. We recognize that labeling it an "Assumption" caused confusion about its generality. We apologize and will **rename/rephrase this condition** (e.g., "Condition 4.1" or similar) and clarify its status in the revised version.
> > It would better to question in Line 66 as "Can SL reduce sampling difficulty in the binary quadratic distributions, as it does in continuous settings, by constructing easily samplable posterior distributions?" since there is various SL methods for discrete sampling tasks as discussed in Appendix A.
>
> This is an excellent suggestion for better framing. We agree and will **revise the question in Line 66** accordingly in the next version.
> > Could you provide the convergence rate/query complexity/iteration complexity explicitly? and compare it with existing results?
>
> Analyzing the convergence rate of the *overall* SL process to the target distribution remains an open and challenging theoretical problem. This difficulty arises from the time-inhomogeneous nature of the process, as discussed in our response to `Reviewer 4aF7's W2`.
>
> We can analyze the operational complexity. Let $N$ be the dimension, $T$ be the number of SL iterations, and $M$ be the total MCMC step budget.
> - **Baseline DMCMC:** Methods like GWG, PAS, DMALA have complexity dominated by operations like gradient/difference calculations or matrix-vector products, typically scaling as $O(MN^2)$.
> - **SL + DMCMC:** SL adds two main steps per iteration: MC estimation (line 118, typically $O(N)$ using posterior samples) and SDE simulation (line 119, $O(N)$). Over $T$ iterations, the total overhead is $O(TN)$. The MCMC sampling itself uses the same total budget $M$, distributed across iterations.
> - **Comparison:** The total complexity for SL+DMCMC is roughly $O(MN^2 + TN)$. In typical high-dimensional settings relevant to MCMC ($N$ large), and practical choices of $T$ (e.g., $T \in ${256, 512, 1024} in our experiments with $M$ up to 10,000), the $O(TN)$ overhead is **negligible** compared to the $O(MN^2)$ cost of the core MCMC sampling.
>
> Thank you for prompting this; we will **add this explicit computational complexity analysis and comparison** to the revised manuscript (likely in the Appendix).

---

### Official Review · Reviewer_eLYf · 2025-03-14

**Overall Recommendation:** 3

**Summary:**

This paper introduces a sampling method for binary quadratic distributions using stochastic localization. It is claimed to be the first theoretical that extends stochastic localization to discrete MCMC samplers. They show polynomial mixing in Glauber dynamics and MH algorithm. Some experiments are provided.

**Claims And Evidence:**

for the most part, as some claims lack empirical justification! For instance:
1. This paper does not provide explicit complexity bounds and only shows polynomial-time mixing but claims that stochastic localization significantly speeds up sampling
2. Stochastic localization does not show a huge improvement in real-world problems given the experiments provided in the paper (0.1-0.5%).
3. This method is not a general purpose framework and can only be applied to binary quadratic distributions and how well it generalizes to other discrete sampling problems is not clear to me.
4. Spectral gap bounds do not show that stochastic localization is computationally faster than other methods.

**Essential References Not Discussed:**

Yes, references are sufficient.

**Experimental Designs Or Analyses:**

It has a variety of benchmarks and multiple MCMC methods are evaluated.good ablation study,  but a few things can be improved:
1. There is no computational cost analysis
2. no comparison to methods such as discrete HMC, Variational inference, etc.
3. This method is heavily dependent on hyperparameters such as \(\alpha \)-schedule and step allocation. the ablation study does not study different temperature settings in MCMC  or the impact of datasets such  as graph sparsity, problem size, etc
4. Since it is claimed that this method outperforms baseline MCMC, the improvement is very small. having some statistical testing to examine the statistical significance can be helpful.

**Methods And Evaluation Criteria:**

1. this paper should provide computational cost analysis and compare it to other MCMC models.
2. No comparison to methods such as discrete HMC
3. Despite the fact that this paper proves polynomial time mixing, there is no clear evidence that it performs better in real world problems and there is no comparison to other MCMC in runtime. Does stochastic localization reduce the number of MCMC steps?

**Other Comments Or Suggestions:**

mentioned in the previous sections.

**Other Strengths And Weaknesses:**

Strengths:
1. It is easy to follow and the use of stochastic localization in a discrete setting is interesting.
2. Applying it to many datasets is interesting and can confirm the theoretical claims.

Weaknesses:
1. While the polynomial mixing has strong theoretical results, the empirical improvements are not drastic.
2. the theoretical assumptions may not hold. For instance, assumption 4.1 requires |h| to grow large which can be violated easily in many real world problems.
3. Comparison is only done  against MCMC based models
4. despite ablation studies, the hyperparameters must be studied more. For instance, do differnet settings change the performance drastically? or is the method robust?

**Questions For Authors:**

1. Could the authors justify how polynomial time mixing implies practical efficiency?
2. How does stochastic localization improve sampling intuitively? What happens if assumption 4.1 fails?
3. Could authors please discuss runtime comparisons?
4, How do results change with different MCCM step allocations? the ablation study seems a bit unclear to me.

**Relation To Broader Scientific Literature:**

This paper extends stochastic localization to BDDs which is built on top of the line of works on stochastic localization for continuous distribution.

**Theoretical Claims:**

1. the proofs look correct but some assumptions (4.1) can be violated easily in many settings. Seems like if the external field is not large enough, the mixing may fail.
2. Theorem 3.1 is mostly based on the literature but it still does not show that stochastic localization is beneficial in discrete setups.
3. The claim that Poincare inequality provides practical efficiency is not quite correct. it does guarantee ergodicity but does not necessarily imply better performance compared to other techniques.

---

> ### Author Rebuttal · Authors · 2025-03-31
>
> Thanks for your valuable feedback. We have consolidated the main points and respond as follows:
> > No complexity bounds of SL and cost analysis comparisons
>
> SL introduces only minimal additional computation compared to MCMC methods. Please refer to our response to ```Reviewer mA2w's last point```.
> > Why can the results about spectral gap and Poincare inequality lead to practical efficiency?
>
> The speed of DMCMC samplers is generally unknown, but SL decomposes the entire DMCMC sampling process into a series of posterior distribution samplings $P(X|Y_i)$ (see Algorithm 1). We prove that sampling from these posterior distributions satisfies Poincaré inequality with high probability, and the existence of Poincare inequality can ensure polynomial mixing, meaning the fast sampling.
> > Satisfiability of assumption 4.1
>
> Please refer to our response to ```Reviewer mA2w's second point```.
> > Theorem 3.1 does not show that SL is beneficial in discrete setups.
>
> Theorem 3.1 describes the convergence of the observation process, which *induces* the SL dynamics (line 130). It provides the theoretical underpinning justifying *why* Assumption 4.1 holds with high probability, thus ensuring the Poincaré inequality and fast posterior mixing are applicable. We will improve the description surrounding Theorem 3.1 in the revision to make this connection clearer.
> >  Extend SL to general setting
>
> While algorithmic extension to general discrete settings is natural by adapting DMCMC samplers, the *theoretical analysis* (proving Poincaré inequalities) becomes significantly more complex (e.g., bounding Dobrushin interdependence matrix for sample space with more than two states). We focused on the binary case to maintain consistency between our algorithms and theoretical guarantees. We acknowledge this limitation and will **add the extension to general discrete distributions as a future work direction** in the conclusion.
> ## Experiment Issues
> > Marginal improvements on the results and statistical confidence
>
> Please refer to our response to ```Reviewer 4aF7's W1``` regarding empirical results.
> > Absence of Discrete HMC baseline and other methods
>
> Thank you for this suggestion. To our knowledge, Discrete HMC is not a standard baseline in the discrete sampling literature or the benchmarks as we referenced. If the reviewer could provide references or implementations suitable for BQDs, we are happy to consider it for future comparisons.
> > Running time and MCMC steps comparison
>
> In our experiments, SL and DMCMC samplers used identical MCMC steps for fair comparison, which results in similar runtimes with overhead $O(TN)$ for SL. We will add concrete runtime comparison results in our revision.
> > Dependence on alpha-schedule, step allocation, sensitivity analysis on hyper-params
>
> In our main results, we use GEOM(2,1) $\alpha$-schedule, exponential decay step allocation, and uniform time discretization for SDE. As detailed in Appendix D.2, we tune:
> - SDE iteration parameters: initial/final noise scale, sample ratio for posterior expectation estimation, number of SDE iterations, and noise level $\sigma$. These parameters are essential for SDE iterations and were also optimized in SLIPS.
> - Two additional parameters for step allocation: decay rate and minimum MCMC steps
>
> Given fixed DMCMC samplers, we believe this represents a minimal set of hyperparameters. Regarding sensitivity to some key hyperparameters:
> - Figure 1 provides comparisons for different step allocation strategies
> - Tables 4-9 present ablation studies for \alpha-schedule, K, and \sigma variations
>
> These results indicate that while performance varies (expected), **SL consistently outperforms baselines across different settings, demonstrating robustness**.
>
> > Impact of temperature settings in MCMC and datasets
>
> We used the *identical* temperature annealing schedule from the DISCS benchmark for *all* methods (Appendix D.1) for fair comparison.
> Regarding dataset characteristics impact, we studied:
> - Graph sparsity effects in Table 1 using ER datasets of different densities, analyzed in line 356 under "Results on MIS"
> - Problem size scaling in Table 3, showing results for different-sized datasets, analyzed in line 380 under "Results on MaxCut"
>
> In all these diverse settings—varying graph densities and problem sizes—**SL consistently outperformed DMCMC across these variations.**
> > Explanation on the ablation on step allocation
>
> Our theory indicates posterior sampling becomes fast as iterations progress, motivating adaptively allocating more MCMC steps to earlier iterations might be better than uniform allocation. Figure 1 tests this: exponential decay allocation (blue bars, more steps early) generally outperforms uniform allocation (orange bars) for the same total MCMC budget, empirically validating the theoretical motivation. We will clarify this explanation in the caption/text.

---

### Official Review · Reviewer_fwS3 · 2025-03-14

**Overall Recommendation:** 3

**Summary:**

The paper proposes a generic localization sampler for binary quadratic distributions. By simulating the observation process similar to [EAM23], the authors propose an unbiased scheme which is capable of sampling from the target faster than a generic MCMC scheme.

**Claims And Evidence:**

The authors provide proofs for their claims as well as some empirical verification. However, the theory for the estimator does seem to be rather preliminary (or perhaps I have not understood it well enough), and I have proposed numerous questions for the authors.

**Essential References Not Discussed:**

As far as I am aware, the most relevant references have been covered.

**Experimental Designs Or Analyses:**

The paper benchmarks their method against the unlocalized samplers on a diverse suite of tasks. The results show convincingly that in practice, the localization scheme can lead to significant speed-ups.

**Methods And Evaluation Criteria:**

Yes, the benchmarks are thorough and the comparison between methods seems fair.

**Other Comments Or Suggestions:**

Line 166: e nhances -> enhances

**Other Strengths And Weaknesses:**

My comments can be found in the other sections, but speaking broadly:

***Strengths***

The empirical results are very convincing.

The spectral gap results are very nice and cover a wide range of algorithms.

***Weaknessess***

The theory is not truly end-to-end and relies on a heuristic assumption. I have asked some clarifying questions in the fields below.

**Questions For Authors:**

My questions are mainly relating to the theory of this work.

1. What are the missing ingredients for proving a bona fide spectral gap almost surely over all iterations, or at least for a sufficiently large tilt?

2. Similar to the question above, can the guarantees of Theorem 1 be made non-asymptotic?

3. Furthermore, if one only has the spectral gap holding with high probability over the randomized measure, what can one infer about the resulting estimator overall for a given iteration budget?

4. Is there any hope of being ``adaptive’’ in this budget, in case the posterior measure is difficult to sample from?

5. How bad is the error from using a Monte Carlo estimator of the posterior mean (what type of error does it induce for the resulting sampler)? Is there any hope for computing closed forms in any of the instances given, similar to the work of El Alaoui et al. for the SK model?

**Relation To Broader Scientific Literature:**

The primary references are discussed; this is an extension of the stochastic localization sampling schemes from [EAM23], which also appeared in earlier/other works [CE23], etc. The paper also engages with a wide array of discrete space MCMC methods.

**Theoretical Claims:**

The authors provide some rigorous spectral gap bounds under a heuristic assumption that the tilt generated by the observation measure is sufficiently large relative to the remaining terms. The proofs under this assumption are rigorous, while the heuristic holds almost surely asymptotically under Theorem 3.1, whose proof is also rigorous.

I skimmed the proofs and they appear correct to me; the final results are also sensible.

---

> ### Author Rebuttal · Authors · 2025-03-31
>
> We are grateful to the reviewer for the thoughtful feedback.
>
> ## Q1
> The core challenge stems from the path-dependent behavior of Brownian motion. Given the observation process $Y_t = \alpha(t)X + \sigma B_t$, we know that $\frac{Y_t}{\alpha(t)} - X = \frac{\sigma B_t}{\alpha(t)}$, which converges to 0 almost surely as $t$ approaches infinity. Unfortunately, establishing uniform convergence of this process presents significant difficulties.
> To guarantee a sufficiently large external field after iterations (which would ensure the spectral gap), we need $|\frac{Y_t}{\alpha(t)}| \geq 1 - \frac{|\sigma B_t|}{\alpha(t)} \geq c$ for some positive constant $c$ after a large time $t \geq T$. This requires establishing upper bounds on $ \frac{|B_t|}{\alpha(t)}$. Analyzing the Law of the Iterated Logarithm reveals that we can only obtain upper bounds on $ \frac{|B_t|}{\alpha(t)}$ after some $T(\omega)$ that depends on the specific sample path $\omega \in \Omega$.
>
> However, by applying Egorov's theorem, it is promising to show that for any $\varepsilon > 0$, there exists a subset $A$ with measure $P(A) < \varepsilon$ such that $ \frac{|B_t|}{\alpha(t)}$ converges to 0 uniformly on $A^c$. This approach could potentially establish high-probability uniform convergence.
>
> ## Q2
> Sure! From line 838:
> > there exists $T$ large enough such that for $t \geq T$, we have $0\leq \Phi\left(\frac{\zeta}{\sigma}-\frac{\alpha(t)}{\sigma\sqrt{t}}\right)\leq\varepsilon$   (Note: We find the typos in lines 833 and 837 and they have been corrected from "$\pm$" symbols to "-". )
>
> this allows us to derive an explicit estimate of $T$ in terms of $\varepsilon$ based on the convergence rate of $\frac{\alpha(t)}{\sigma\sqrt{t}}$ and properties of the normal CDF.
>
>
> ## Q3
> In practice, the algorithm proceeds with a *specific* realization $Y_t = y_t$. Theorem 3.1 guarantees that for a sufficiently large $t$, the generated $y_t$ will induce a large external field with high probability. Consequently, the posterior distribution $q_t(X|Y_t=y_t)$ that we actually sample from using DMCMC will satisfy the conditions for the Poincaré inequality (Assumption 4.1) **with high probability**. This means that the DMCMC sampler used for the posterior estimation will mix polynomially fast **with high probability** for sufficiently large $t$.
>
> ## Q4
> Your intuition is correct. Our theory suggests posterior sampling becomes easier over iterations $t$. This motivates allocating the MCMC budget adaptively. In Section 6.2, under "Impact of MCMC Step Allocation," we empirically validate this. We demonstrate that an adaptive allocation strategy (specifically, exponential decay allocation) **achieves superior performance** compared to a uniform allocation across the vast majority of cases (see Figure 1), confirming the practical benefit of adapting the budget.
>
> ## Q5
> Sure. Since DMCMC is an irreducible reversible Markov chain with finite state space {$-1,1$}$^N$, satisfying the conditions of Theorem 1.1 in [1], we can derive an error estimate for the estimator of the mean of $\nu_{\beta,h}$ that depends on the spectral gap:
>
> $$P_{q}\left[\left|\left|\frac{1}{n}\sum_{i=1}^nX_i-E_{\nu_{\beta,h}} [X]\right|\right|<\varepsilon\right]\geq1-2e^{\gamma_{gap}/5}N_q\exp\left(-\frac{n\varepsilon^2\gamma_{gap}}{4Var_{\nu_{\beta,h}}[X]\cdot(1+g(5\varepsilon/Var_{\nu_{\beta,h}}[X]))}\right),$$
>
> where $X_1\sim q$, $N_q=\|\|\frac{q}{\nu_{\beta,h}}\|\|_2$ measures the error from sampling before the Markov chain reaches stationarity, and function $g(x)=\frac{1}{2}\left(\sqrt{1+x}-(1-x/2)\right)$.
>
> Thank you for your insightful comment, we will add a discussion of this bound to the Appendix in the revision.
>
> ---
>
> [1] Lezaud P. Chernoff-type bound for finite Markov chains[J]. Annals of Applied Probability, 1998: 849-867.

---

> > ### Comment · Reviewer_fwS3 · 2025-04-04
> >
> > I thank the authors for their thorough response. I am maintaining my score, although I would be happy to see the theoretical results improved to avoid dependence on Assumption 4.1.

---

> > > ### Author Response · Authors · 2025-04-04
> > >
> > > Thank you for your response. We sincerely apologize for the misunderstanding regarding 'Assumption 4.1'—it should instead be termed a '**Fact**' rather than an 'Assumption'. In the SL framework, 4.1 holds as |h| (In our algorithm, h-value should equal $b+\frac{\alpha(t)Y_t}{\sigma^2t}$) grows infinitely with iterations (line 161 on the right column), ensuring satisfaction after sufficient steps with high-probability. We acknowledge that labeling this as an "assumption" was potentially misleading and will revise this in our manuscript.
> > >
> > > We hope this clarification could lead to an positive influence on the score evaluation. Please let us know if you have additional concerns regarding the content. We appreciate your constructive comments and look forward to addressing any further questions to improve our work.

---

### Official Review · Reviewer_4aF7 · 2025-03-14

**Overall Recommendation:** 3

**Summary:**

This paper studies stochastic localization for sampling from binary quadratic distributions.
As a main theoretical contribution, the authors prove Poincare inequalities for the sampling procedure from the (discrete) posterior distribution \\(q_t(x \mid y)\\) in stochastic localization, and thus establish the convergence rate of the posterior sampling step.

A key implication is that the spectral gap for posterior distribution sampling increases as the time \\(t \to +\infty\\).
This is achieved by an asymptotic argument.
As a result, posterior distribution sampling becomes easier and easier in the later stage of stochastic localization.

**Claims And Evidence:**

Yes, the claims are generally well-supported by evidences.
I like the ablation study in Section 6.2 that verifies the intuition that posterior sampling becomes easier as \\(t \to +\infty\\).

**Essential References Not Discussed:**

Not that I am aware of.

**Experimental Designs Or Analyses:**

Yes, I read through the experiment section; see comments in Other Strengths And Weaknesses.

**Methods And Evaluation Criteria:**

Yes, the methods and evaluation criteria make sense.

**Other Comments Or Suggestions:**

Line 166: "e nhances" -> "enhances"

**Other Strengths And Weaknesses:**

1. I think the empirical results is not strong enough.
The experiments suggest that using stochastic localization for discrete distributions is better than directly using a discrete MCMC method.
However, the performance gaps between them are tiny (I am assuming \\(\pm\\)xxx in the tables represents one standard deviations).
Indeed, the baseline performances are always within one standard deviation of stochastic localization.

1. Another weakness is that this paper actually does not prove how fast the samples produced by stochastic localization converge to the target distributions.
The results in this paper only apply to the posterior sampling---an intermediate step in stochastic localization.

**Questions For Authors:**

None at this moment.

**Relation To Broader Scientific Literature:**

This paper extends previous stochastic localization methods by El Alaoui & Montanari (2022) and Grenioux et al. (2024) to discrete distributions.

**Theoretical Claims:**

I think the theoretical claims make sense.
I have not checked their proofs, though.

---

> ### Author Rebuttal · Authors · 2025-03-31
>
> We sincerely thank the reviewer for these insightful comments.
>
> ## W1
> From an experimental perspective, the absolute improvements are indeed modest. However, we would like to emphasize two key points:
> 1.  **Strong Baselines:** Our comparison is between standard DMCMC and SL combined with the *same* DMCMC sampler (SL+DMCMC). The base DMCMC samplers used are already highly effective, demonstrating strong performance on their own. As shown in Table 3, all DMCMC baselines significantly outperform the commercial solver Gurobi, indicating they are already operating near optimal performance levels for these challenging tasks. The fact that SL consistently provides further improvements, even on top of these powerful, well-established DMCMC methods, highlights its efficacy.
> 2.  **Benchmark Standards:** Our experimental setup, comparison metrics, and implementation strictly follow the established DISCS benchmark [1]. As evidenced by Tables 5-7 in [1], even state-of-the-art DMCMC samplers often exhibit performance differences within one standard deviation of each other. Substantial performance overlap when considering one standard deviation is common in this domain. Therefore, comparing methods based on mean results is standard practice, as adopted in [1] and our work.
>
> Thanks for highlighting this point. In the revised version, we will **add text to better characterize the near-optimality of these baselines** to properly contextualize SL's performance improvements and clarify the standard deviation reporting convention.
>
> [1] Goshvadi K, Sun H, Liu X, et al. DISCS: a benchmark for discrete sampling[J]. Advances in Neural Information Processing Systems, 2023, 36: 79035-79066.
>
> ## W2
> This is a profound but challenging problem. You are correct that our current theoretical results guarantee fast mixing (via Poincaré inequality) for the *posterior sampling step* within SL, but not the convergence rate of the *overall SL process* to the final target distribution. Analyzing the full SL process is difficult because $\frac{Y_{t}}{\alpha(t)}=X+\frac{\sigma B_{t}}{\alpha(t)}$ is not a time-homogeneous Markov process, making standard convergence analysis tools like functional inequalities hard to apply directly to the overall dynamics.
>
> One promising direction, as you note, involves analyzing the properties of $\frac{Y_{t}}{\alpha(t)}$. Since these random variables have densities with explicit expressions, calculating the KL-divergence between distributions at different times $t_1$ and $t_2$ might be feasible. This could potentially reduce the problem to estimating the ratio of partition functions of BQDs under varying external fields. A more general framework, perhaps using Wasserstein distance, might also be needed to characterize the convergence of the overall SL process. We acknowledge this limitation and consider establishing the convergence rate of the full SL sampler a key direction for **future work, which we will explicitly mention in the conclusion**.
>
> *(Typos addressed)*

---

### Decision · Program_Chairs · 2025-05-01

**Decision:**

Accept (poster)

**Comment:**

The authors study applying stochastic localization methods to binary quadratic distributions.  Their main result is a proof of Poincare inequalities for the sampling procedures used to estimate the expectations used in the SL procedure.  They show that this sampling step becomes easier as the SL process continues and propose a schedule to take advantage of this property, although as noted by Reviewer 4aF7, their results do not extend to providing information about the convergence of the overall process.  Experimental comparisons show pretty minor but consistent improvement over each sampling technique without SL.  Reviewers were generally positive, with some concerns about the impact of the theoretical contribution (narrow setting; results are not directly about the quality of the overall sampling procedure, experimental impact, and some presentation (connections to SL used in other discrete settings, etc.).